# A Bayesian network approach incorporating imputation of missing data enables exploratory analysis of complex causal biological relationships

**Richard Howey**[1], **Alexander D. Clark**[2], **Najib Naamane**[2], **Louise N. Reynard**[3], **Arthur G. Pratt**[2,4], **Heather J. Cordell**[1] *

**1** Population Health Sciences Institute, Newcastle University, Newcastle upon Tyne, United Kingdom,
**2** Translational and Clinical Research Institute, Newcastle University, Newcastle upon Tyne, United Kingdom,
**3** Biosciences Institute, Newcastle University, Newcastle upon Tyne, United Kingdom, **4** Musculoskeletal Services Directorate, Newcastle upon Tyne Hospitals NHS Foundation Trust, Newcastle upon Tyne, United Kingdom

* heather.cordell@newcastle.ac.uk

**Data Availability Statement:** All relevant simulated data are within the manuscript and its Supporting information files. Scripts underlying the computer

## Abstract

Bayesian networks can be used to identify possible causal relationships between variables based on their conditional dependencies and independencies, which can be particularly useful in complex biological scenarios with many measured variables. Here we propose two improvements to an existing method for Bayesian network analysis, designed to increase the power to detect potential causal relationships between variables (including potentially a mixture of both discrete and continuous variables). Our first improvement relates to the treatment of missing data. When there is missing data, the standard approach is to remove every individual with any missing data before performing analysis. This can be wasteful and undesirable when there are many individuals with missing data, perhaps with only one or a few variables missing. This motivates the use of imputation. We present a new imputation method that uses a version of nearest neighbour imputation, whereby missing data from one individual is replaced with data from another individual, their nearest neighbour. For each individual with missing data, the subsets of variables to be used to select the nearest neighbour are chosen by sampling without replacement the complete data and estimating a best fit Bayesian network. We show that this approach leads to marked improvements in the recall and precision of directed edges in the final network identified, and we illustrate the approach through application to data from a recent study investigating the causal relationship between methylation and gene expression in early inflammatory arthritis patients. We also describe a second improvement in the form of a pseudo-Bayesian approach for upweighting certain network edges, which can be useful when there is prior evidence concerning their directions.

simulations and the BN and CIT analysis of the processed DNA methylation and gene expression data from Clark et al. (2020) may be found at https://github.com/drhowey/BayesNetty/blob/main/docs/scripts.zip. The original DNA methylation data from CD4 and B cells used in this study, together with paired transcriptome data, are available in the GeneExpression Omnibus database (accession no. GSE137634; http://www.ncbi.nlm.nih.gov/geo). Annotated scripts used for the processing and CIT analysis of these data as performed by Clark et al. (2020), together with the input file used for the current work, may be found at https://github.com/aclark5/Lymphocyte_meQTL.

**Funding:** This research was funded in whole, or in part, by the Wellcome Trust [Grant numbers 102858/Z/13/Z and 219424/Z/19/Z] (RH, HJC). For the purpose of open access, the author has applied a CC BY public copyright licence to any Author Accepted Manuscript version arising from this submission. This work was also supported by a grant from JGW Patterson Foundation (https://jgwpattersonfoundation.co.uk/) 30015.088.036/P/IXS (AGP). The funders had no role in study design, data collection, analysis, decision to publish, or preparation of the manuscript.

**Competing interests:** The authors have declared that no competing interests exist.

## Author summary

Data analysis using Bayesian networks can help identify possible causal relationships between measured biological variables. Here we propose two improvements to an existing method for Bayesian network analysis. Our first improvement relates to the treatment of missing data. When there is missing data, the standard approach is to remove every individual with any missing data before performing analysis, even if only one or a few variables are missing. This is undesirable as it can reduce the ability of the approach to infer correct relationships. We propose a new method to instead fill in (impute) the missing data prior to analysis. We show through computer simulations that our method improves the reliability of the results obtained, and we illustrate the proposed approach by applying it to data from a recent study in early inflammatory arthritis. We also describe a second improvement involving the upweighting of certain network edges, which can be useful when there is prior evidence concerning their directions.

## Introduction

Genome-wide association studies (GWAS) have had considerable success in detecting genetic variants (typically single nucleotide polymorphisms, SNPs) associated with phenotypic outcomes. There is now considerable interest in using integrative analysis with additional data types (such as measures of gene expression, DNA methylation or protein levels) to better understand the biological mechanisms underpinning these results. One possible analysis approach is to use Bayesian Networks (BNs), whereby potential causal relationships between many different genetic, biological and phenotypic variables may be explored, taking advantage of the fact that genetic variables can act as instruments to help orient the directions of relationships between other variables. This approach has been shown to perform competitively with other causal inference methods [1] and in some cases to even have advantages over competing approaches such as Mendelian Randomisation (MR) and its extensions [2].

BNs were first formalized and developed by Pearl [3, 4] in relation to encoding expert knowledge in the context of artificial intelligence (AI), and they have now become widely applied in the social and natural sciences. A BN comprises a graphical model known as a directed acyclic graph (DAG) and an accompanying joint probability which describes the conditional dependencies of the variables [5]. Variables are represented by nodes and their conditional relationships by directed edges (arrows). The terms "node" and "variable" are used interchangeably throughout this manuscript. The joint probability of the DAG is decomposed as a product of local probabilities where the local probability of each "child" variable is determined by its conditional dependencies on the "parental" variables immediately directed towards it [6]. The local probability distributions can be defined in many ways, but a popular approach, which we use throughout, is for discrete variables to take a multinomial distribution and continuous variables to take a multivariate normal distribution.

In recent years it has become popular [7, 8] to emphasize that, strictly speaking, BNs are defined purely in terms of their encoding of conditional independence relationships between variables, without any implication that these should represent causal relationships. Effectively, a Bayesian network can be considered to be "nothing more than a compact representation of a huge probability table" [8]. However, it is not clear that this is how they were originally envisaged. Pearl (1985) [3] stated "Bayes Networks are directed acyclic graphs in which the nodes represent propositions (or variables), the arcs signify the existence of direct *causal influences* between the linked propositions", while Pearl (1988) [4] stated "a Bayesian network is a

directed acyclic graph whose arrows represent *causal influences* or class-property relationships" and "Bayesian networks are DAGs in which the nodes represent variables, the arcs signify the existence of direct *causal influences* between the linked variables" (our emphasis added). As pointed out by Heckerman et al. [9], "in practice, Bayesian networks are typically constructed using notions of cause and effect" and "learning a network structure is useful, because we can sometimes use structure to infer causal relationships in a domain, and consequently predict the effects of interventions". Di Zio et al. [10] state, in relation to using BNs for imputation, "The direction of the edges is usually interpreted as a causal relationship between the two variables", although they go on to point out that "this interpretation sometimes is quite severe" and "is not necessary for our purposes".

Effectively, the difference between an acausal BN and a causal BN is in the meaning and interpretation ascribed to it: in an acausal BN, the directed arcs or arrows between variables represent purely conditional independence relationships, while in a causal BN they represent direct functional relationships among the corresponding variables [11], such that an intervention on (or manipulation of) the value of a parent variable will result in a corresponding change in the probability distribution of a child variable [5]. Heckerman [12] uses the terminology "responsiveness" to encapsulate this idea of the probability distribution for one variable being determined by (or responsive to changes in) the value of another variable. We note that this definition of causality (in terms of the *probability distribution* of one variable being determined by, or responsive to, the value of another) is a population-level concept that differs from the concept of *individual-level* chains of causation [8], which can be potentially identified using methods that operate on more general causal diagrams, such as structural causal models [11, 13].

As pointed out by Pearl and Mackenzie [8], all the probabilistic properties of BNs remain valid in causal diagrams, and Pearl (2009) [11] notes that "Probabilistic relationships, such as marginal and conditional independencies, may be helpful in hypothesizing initial causal structures from uncontrolled observations". To the extent that a causal diagram *implies* a set of probabilistic relationships, and a BN *encodes* a set of probabilistic relationships, we consider that the search for BNs that are well-supported by observed data can also be considered as a search for potential causal relationships that are well supported by the observed data. Spirtes [5] points out that "In order to use samples from probability densities to make causal inferences, some assumptions relating causal relations to probability densities need to be made". The reasonableness of (and/or justification for) such assumptions is likely to be domain specific, but the three assumptions most typically made [5, 11] are 1) the causal Markov assumption, 2) the causal faithfulness assumption, and 3) the causal sufficiency assumption. The causal Markov assumption states that a variable is independent of all other variables, except for its effect or descendent ("child"/"grandchild" etc.) variables, conditional on its direct causal (or "parent") variables [5, 6, 14]. The causal faithfulness assumption (also known as "stability" [11]) states that the network structure and the causal Markov relations assumed represent all (and the only existing) conditional independence relationships among variables [5, 15]. The causal sufficiency assumption (which actually follows from the first two assumptions [7]) corresponds to asserting there are no external variables which are causes of two or more variables within the model, implying that all causes of the variables are included in the data and there are no unobserved confounding variables [5, 15, 16].

A variety of different algorithms have been proposed to search for causal models and/or BNs that are well-supported by a given set of data. These include constraint-based methods such as the PC algorithm [17] and the Fast Causal Inference (FCI) algorithm [18], and score-based methods, which can include both Bayesian [9] and frequentist approaches. (Note that the term "Bayesian" within BNs does not imply the use of a Bayesian—as opposed to a

frequentist—paradigm, but rather refers to the fact that certain calculations rely heavily on Bayes' theorem). Constraint-based methods generally start with a fully connected graph and carry out a series of marginal and conditional independence tests to decide which edges to remove. This approach can be considered non-parametric, as it focuses on testing conditional independencies rather than requiring specification of a parametric likelihood. Score-based methods require a likelihood and are thus parametric; the idea is to move around through network space in order to determine the most plausible BN(s) (whose structure and parameter values are most compatible with the observed data), i.e the BN(s) with the best score (highest or lowest, depending on how the score function is defined), or the highest posterior probability, out of all possible BNs. BNs that represent the same set of conditional independence relationships between the variables have the same score and cannot be distinguished from one another; they are said to be observationally equivalent [11] and, in the absence of any other information to choose between them, may be considered equally plausible.

One advantage of BNs is their ability to model large complex data sets in a flexible manner. This is useful when modelling many causal relationships in "omics scale" biological data sets [19], such as in studies of DNA methylation and gene expression [20] or metabolites [21]. However, this advantage does have a cost in terms of the large computational power required to find the best fit network when there are many variables. As the number of variables increases, the number of potential networks increases super-exponentially, and therefore it is necessary to search through network space rather than evaluate every possibility. Also, to fit a network it is necessary to have individual-level data rather than summary data, which is a disadvantage compared to some other causal inference methods such as MR.

A weakness of almost any statistical analysis is how missing data is handled, and for many analyses this simply amounts to deleting any observations that have any variables with missing data. This is the usual approach with BNs. Here we present a new data imputation method designed to handle missing data in order to improve identification of the best fit network (or networks) in the context of BNs. We note that this is a different goal from trying to obtain the most accurate imputation of missing data; although one might hope that accurate imputation of missing data would result in accurate identification of the best fit network, it is the latter (accurate identification of the best fit network) rather than the former (accurate imputation of missing data) that is our primary goal. This is a non-trivial task as the overall structure of the data must be maintained so that false relationships are not induced between variables. We present a fast, efficient, user-friendly implementation of our method with freely available open source code (and working examples). Further speed-ups may be achieved by the use of parallel computing. Our software package, BayesNetty [2, 22] uses the (frequentist) score-based algorithm from the R package `bnlearn` [7] as a basis, which is then extended to encompass our imputation approach. We also propose a novel addition that takes advantage of existing knowledge about relationships between variables to define soft constraints, whereby the directions of certain edges are up- or down- weighted through the assignment of prior probabilities, in a pseudo-Bayesian approach.

Although the primary envisaged aim of our software is for application to genetic and associated multi-omics data, in principle it may be applied to any suitable data types, and we include demonstration of our imputation method to several non-genetic data sets. An advantage of using genetic data in this type of analysis is the fact that genes are assigned at birth, and so can act as "causal anchors", whereby arrows (interpreted as causal relationships) should only be directed outwards from genetic variables towards the non-genetic variables that they influence. This constraint provides a practical way to choose between BNs that are observationally equivalent; if an acausal BN is supposed to reflect the independencies implied by the corresponding causal BN, then it makes sense to only consider BNs that obey the desired constraint. Our

software package, BayesNetty, specifically enforces this constraint by only considering BNs that have arrows coming out from (rather than going into) any variables that have been labelled as genetic variables. In addition, it is possible within BayesNetty to "blacklist" certain arrows such that edges between specified nodes are not allowed to exist, effectively constraining the direction that any arrow between these nodes can take.

Previous work has used BNs to impute missing values in data sets when the network is known (or has been estimated) by using the resulting probability distribution for child nodes, conditional on their parental nodes, to substitute in the expected values [10, 23]. Our imputation method operates very differently from these approaches; we use an estimated BN to select individuals that are subsequently used to perform "nearest neighbour" imputation [24], whereby the missing value of a variable in one (target) individual is substituted by the observed value from another individual that is deemed sufficiently "similar" to the target individual. Our method has some similarities with an approach proposed by Miyakoshi and Kato [25] which uses the weighted contribution of $k$ nearest neighbours (chosen based on an estimated BN) to inform the imputation of a target individual. However, Miyakoshi and Kato focus primarily on (a) assessing imputation accuracy and (b) using the resultant complete data set for classification, whereas we propose using the resultant complete data set for carrying out further BN analysis.

There also exist methods that aim to learn the structure of a BN while accounting for missing data [26–30] but, again, most only handle discrete data. As far as we are aware, ours is the first method that can handle general mixed discrete/continuous data. Our main interest is in continuous data for application to genetic and multi-omics data sets, which thus forms the focus of the examples in this manuscript, but we do also consider discrete variables in our applications to early inflammatory arthritis and to a couple of benchmark discrete BN data sets.

## Results

### Computer simulations to explore the performance of our proposed imputation approach

**Imputation of small networks (3 or 5 variables respectively).**   We start by using computer simulations (see S1 Text) to investigate the performance of our imputation algorithm (along with other approaches) when applied to small, very simple networks. A larger network will, in effect, be composed of many smaller sub-network structures, and so it is of interest initially to see how well our approach performs without the support of the extra information that may be acquired from imputing variables elsewhere in a larger network.

Fig 1 shows the proportion of best fit networks that correctly identified the network structure used in the simulation model (or an observationally equivalent network) when there are only three variables, one of which has 90% missing data. (See S1 Text for details of the simulation models). Observationally equivalent networks, representing the same dependencies/independencies between variables, cannot be distinguished and are all considered as "correct". We compare results from our two imputation approaches—our default algorithm ("Imputed") and the imputation with complete training data ("Imputed CT") algorithm (see Methods)— with those obtained (i) when there is no missing data ("Full") (this could also be described as a "complete case" scenario), (ii) when any observations with missing data are removed ("Reduced"), (iii) when missing data is replaced by random values drawn from the same variable ("Random"), (iv) when the missing values are imputed using the expectation-maximisation (EM) algorithm implemented in the bnlearn R software package [26], (v) using simple mean imputation, (vi) using our default imputation approach but with all variables (rather

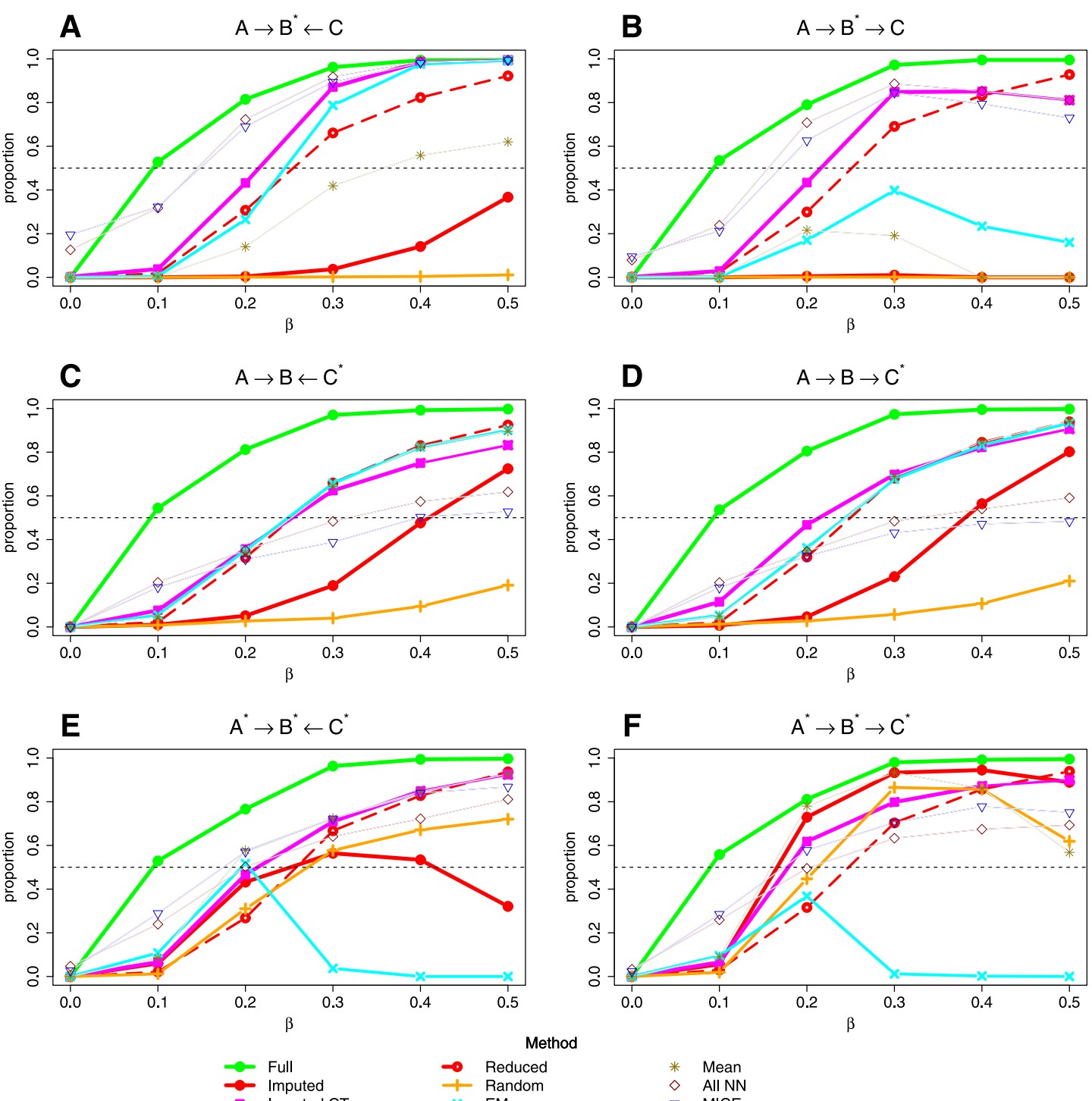

**Fig 1. Line graphs showing the proportion of simulation replicates in which BayesNetty (and alternative approaches) recovered the simulation network (or an observationally equivalent network) for different data sets.** The title of each plot shows the simulating model and the asterisk denotes which variable or variables are missing in 1800 missing individuals from a total of 2000 individuals (only one variable is missing for each individual). The parameter $\beta$ denotes the model strength that was used to simulate the data. Full: the data consisted of the original simulation data with no missing values. Imputed: the data was imputed using our default algorithm. Reduced: the data was reduced to the 200 individuals with no missing values. Imputed CT: the data was imputed with complete data training. EM: the data was imputed with an expectation-maximisation algorithm in R. Random: missing data was replaced by random values drawn from the same variable. Mean: missing data was replaced by the mean value of the same variable. All NN: the data was imputed using our default algorithm but with all variables used to inform the choice of nearest neighbour. MICE: the data was imputed using the multivariate imputation by chained equations approach.

than a selected set as used in our default and CT algorithms) used to inform the choice of nearest neighbour ("All NN"), and (vii) using the multivariate imputation by chained equations (MICE) approach [31] for imputing missing data.

Fig 1 illustrates that the Imputed CT algorithm is generally effective at correctly identifying the network structure, showing either similar power or slightly outperforming the strategy of using the reduced data set. The EM algorithm is also quite effective for simulation scenarios (A), (C) and (D) but quite poor for simulation scenarios (B), (E) and (F). Replacing missing data by random values drawn from the same variable does not perform well. Mean imputation works well in scenarios (C)-(F) but very poorly in scenarios (A) and (B). The "All NN" and MICE methods outperform the Imputed CT algorithm in scenarios (A) and (B) but do not perform as well in scenarios (C), (D) and (F). The default imputation algorithm does not perform well when only one variable is missing, simulation scenarios (A)-(D), but performs better when the missing data is spread between different variables as seen in simulation scenarios (E) and (F); our default imputation method is really designed for larger networks ideally with sparser missing data. In larger networks, when missing data is replaced with random data (which forms the first step of our default imputation algorithm), we can obtain a fairly accurate initial BN, which leads to better overall performance, as demonstrated in the following paragraphs.

Fig 2 shows recall and precision for the 5 variable model, $A \rightarrow B \leftarrow C \rightarrow D \leftarrow E$, where either $B$ and $D$ or $A$ and $E$ (indicated by asterisks) have a complex missing data pattern (see S1 Text). Recall (also known as sensitivity) is the proportion of true edges that were actually retrieved in the best fit network, while precision (also called positive predictive value) is the proportion of true edges among the retrieved edges. Our default imputation method shows overall the highest recall and precision, considerably outperforming using the reduced data set and also outperforming the "All NN", MICE and Imputed CT algorithms. When variables $A$ and $E$ are the ones with missing data, the mean imputation, "Random" and "EM" methods also perform quite well, but their performance is not quite as good as either our default imputation method or the Imputed CT algorithm when variables $B$ and $D$ are the ones with missing data. Results for a different 5 variable model, $A \rightarrow B \rightarrow C \rightarrow D \rightarrow E$, are shown in S1 Fig, with broadly similar conclusions.

**Imputation of networks from the Bayesian Network Repository.** We also applied the same methods to one continuous and two discrete data simulation networks taken from the `bnlearn` BN repository, whose structure and generating parameters are known, containing 46, 37 and 27 nodes respectively (see S1 Text). S2 Fig shows recall and precision for the best fit network, where the percentage of missing data in 450 of the 500 individuals is given on the horizontal axis (except for the green lines, which relate to analysing the full data and so illustrate the maximum value achievable). Our default imputation method shows a clear improvement over using the reduced data set (with the improvement reducing as the proportion of missing data increases), with the Imputed CT algorithm seen to be just slightly less effective. The EM algorithm also shows an improvement over using the reduced data set except for the "ecoli70" network when there is 30% or 40% missing data. However, it is clearly less effective than our own imputation method except perhaps for the "alarm" network with 40% missing data. The other imputation methods considered (mean imputation, "All NN" and MICE) show similar recall but lower precision than our approaches for the continuous data set, "ecoli70". For the other two discrete data sets, where mean imputation does not apply, "All NN" and MICE outperform our proposed approaches. This suggests that, for purely discrete data sets, our approaches may not be strictly optimal, although our main concern is with continuous data.

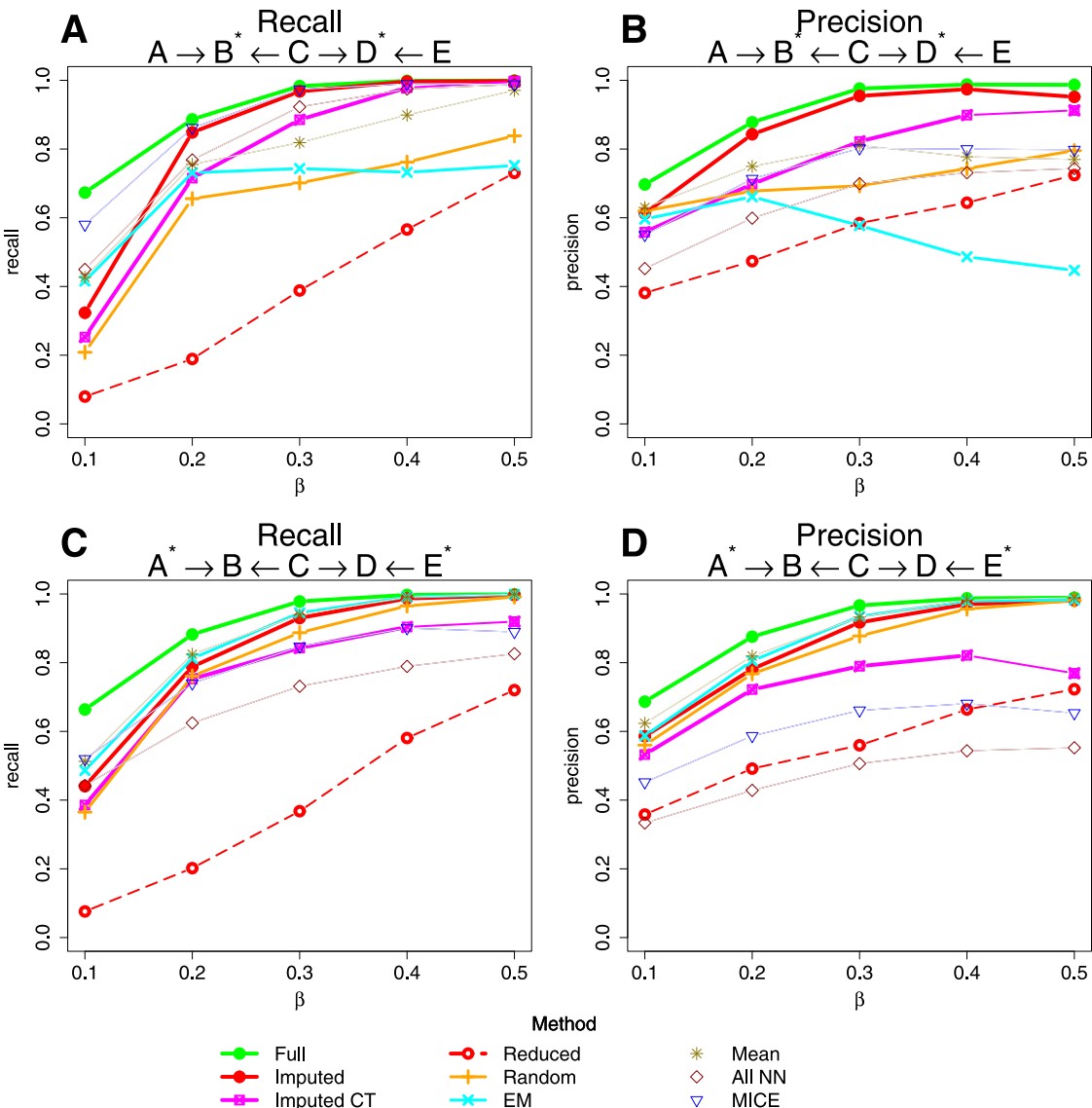

**Fig 2. Line graphs showing the recall and precision of the best fit network found by BayesNetty (and alternative approaches) when the simulation network was $A \rightarrow B \leftarrow C \rightarrow D \leftarrow E$ for different data sets.** The asterisks denote which variables had missing values for 1800 individuals from a total of 2000 individuals. The parameter $\beta$ denotes the model strength that was used to simulate the data. Full: the data consisted of the original simulation data with no missing values. Imputed: the data was imputed using our default algorithm. Reduced: the data was reduced to the 200 individuals with no missing values. Imputed CT: the data was imputed with complete data training. EM: the data was imputed with an expectation-maximisation algorithm in R. Random: missing data was replaced by random values drawn from the same variable. Mean: missing data was replaced by the mean value of the same variable. All NN: the data was imputed using our default algorithm but with all variables used to inform the choice of nearest neighbour. MICE: the data was imputed using the multivariate imputation by chained equations approach.

**Imputation of networks with many variables.**   We next simulated data from a network with 31 variables (see Fig 3A), where the 10 expression data variables, prefixed with "ex", have data values set to missing with a 20% probability. The missing data were filled in either as a random sample from the observed data (see S1 Text) or using our imputation approaches. It was not possible to use the EM implementation in the `bnlearn` R software package in this case, as it does not deal with mixed (discrete and continuous) data. (BayesNetty can deal with

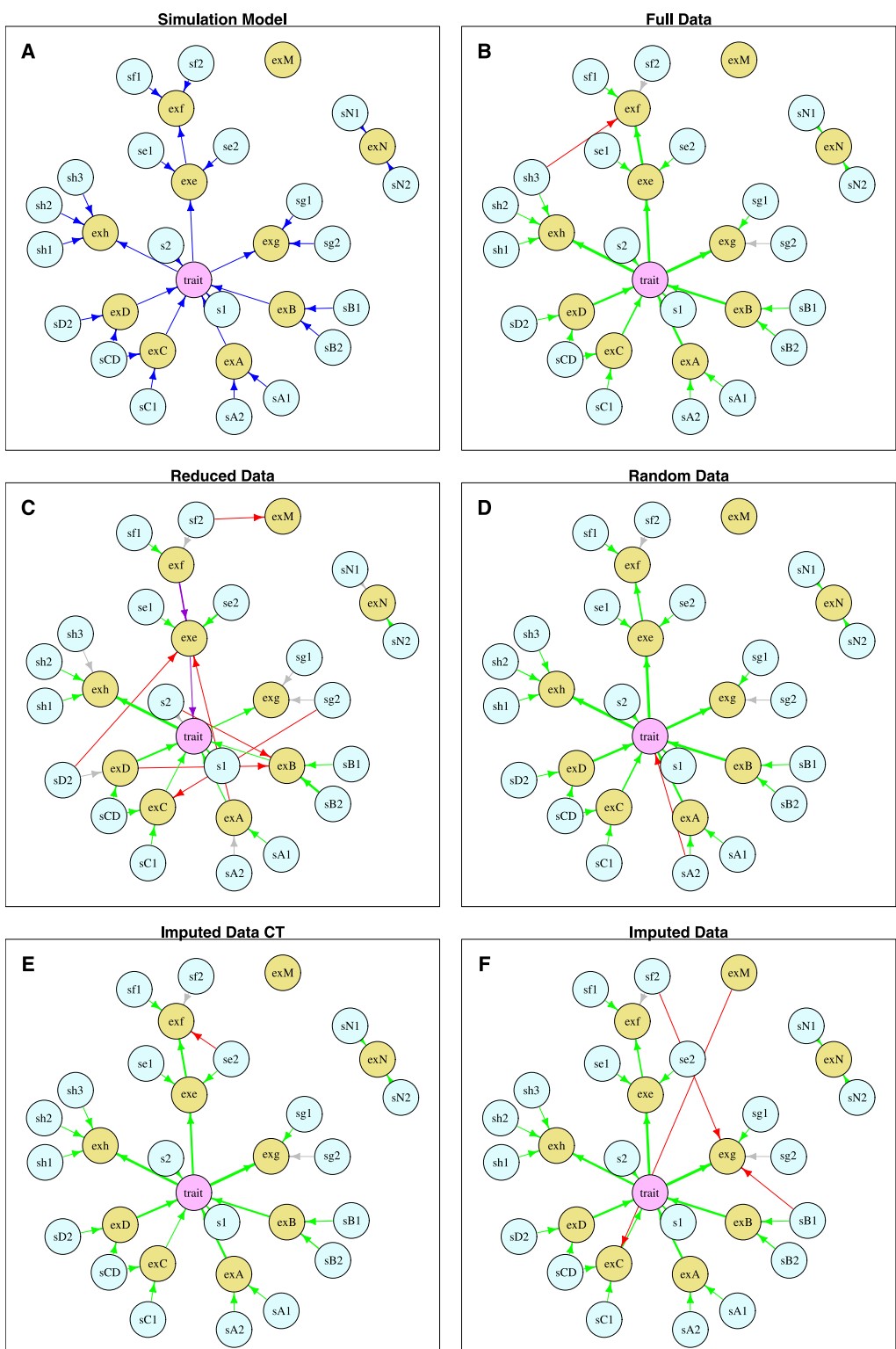

**Fig 3. Graphs showing the networks for (A) the simulation network; (B) the best fit network using the full data with no missing values; (C) the best fit network using data reduced to the individuals with no missing values; (D) the best fit network using randomly imputed data; (E) the best fit network using imputed data (imputed CT algorithm); (F) the best fit network using imputed data (default algorithm).** Graphs (B)-(F) show best fit networks for one typical data simulation when $\beta = 0.3$. The green arrows show edges that are present in the simulation network,

red arrows edges that are not present, purple arrows edges that are in the reverse direction and light grey (almost invisible) arrows edges that are missing in the best fit network. Light blue nodes represent genetic factors, yellow nodes represent gene expression measures and the pink node represents the trait. The thicknesses of the arrows are scaled to show the significance of the edges (between a minimum and maximum thickness), based on a $\chi^2$ value for each edge, which is obtained by removing the edge and comparing the log likelihoods of the network scores with and without the edge using a likelihood-ratio test.

mixed data, although in this case we chose to code the discrete genetic variables as continuous allele dosages, however the EM implementation in the `bnlearn` R software package did not permit this functionality).

Fig 3(B)–3(F) show the best fit networks for one typical data simulation when the network strength parameter, $\beta$ (see S1 Text), was set to 0.3. Fig 3B shows the best fit network for the full data, which recovers most of the edges and adds one incorrect edge, whereas the best fit network using the reduced data (Fig 3C) is missing many edges, adds six incorrect edges and two edges are in the incorrect direction. Fig 3(D)–3(F) show the best fit network using either randomly imputed data or our imputation approaches. All three methods are successful in detecting most true edges with the identification of a few incorrect edges. The recall and precision of directed edges of the best fit BN over 100 simulation replicates achieved using our methods (in comparison to other approaches) are shown in Fig 4. It can be seen that there is a substantial increase in both recall and precision when using imputation compared to using the reduced data. The default imputation algorithm and the Imputed CT algorithm perform fairly similarly, with the default algorithm showing a slight advantage with respect to recall and the Imputed CT algorithm showing a slight advantage with respect to precision. The recall and precision for either randomly replaced or mean imputed data is also substantially better than when using the reduced data, although the precision decreases slightly for large values of $\beta$. This shows the benefit of recovering non-missing data from individuals that have missing data even by simple methods for replacement of missing values. The "All NN" and MICE methods also perform well in terms of recall, but show less good performance in terms of precision, compared to the other imputation methods considered.

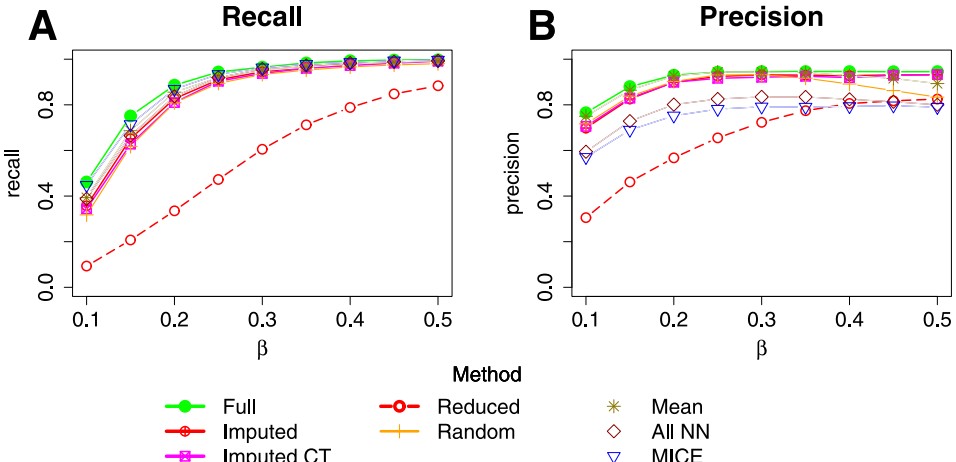

**Fig 4. Line graphs showing the recall and precision of the best fit network found by BayesNetty and alternative approaches when the simulation network is set to the one shown in Fig 3A for different data sets.** Expression data was set to be missing with a 20% probability. The parameter $\beta$ denotes the model strength that was used to simulate the data.

**Utility of applying soft constraints.** We next investigated the utility of using prior knowledge (see S1 Text) to improve detection of certain directed edges in a network. The rationale for including this option was the idea that it might improve power in the situation where the existence of such directed edges is well-supported by prior evidence (or scientific theory), without seriously biasing the results if the belief turned out to be misplaced (as the prior would be overcome by the data). We note that methods for assigning such priors are likely to be context/application specific, although there exists a literature on expert elicitation of Bayesian priors [32].

We start with a simple investigation of varying the prior probabilities of edges within a 3-variable or 4-variable network. See S1 Text, S3 and S4 Figs and the S4 Fig legend for full details.

We also used a simulation from a more complex network (Fig 5A) to demonstrate the effects of varying the prior probabilities of some edges on the ability to detect other edges that were not assigned prior probabilities in the best fit models. Fig 5B shows the proportion of times each edge is detected for different assumed prior probabilities of the red edges (labelled with $p$). Naturally the power for the red edges is affected the most, whereby higher prior probabilities result in higher probabilities of detection. The power for detection of the cyan edges (labelled with x) also increases when the prior probabilities of the red edges are increased, since, when the red edges are orientated in the correct direction, it helps to better resolve the correct direction for the cyan edges as they are connected to the same nodes as the red edges. When the prior probabilities of the red edges are set to less than 0.5 (so in the incorrect direction), the proportion of the cyan edges detected decreases, but not by too much as there are other edges in the network that aid the resolution of the correct direction. The blue edges (labelled by +) also have increased detection probabilities when the prior probabilities of the red edges are increased, but not to the same extent as the cyan edges, as there are fewer other edges connected to these edges. The detection proportion of the blue edges also decreases by more than the cyan edges when the prior probability of the red edges is in the incorrect direction. The detection proportion of the black (labelled with o) and green (labelled with ∼) edges are not much affected due to their weak simulated effect sizes. The overall average recall and precision is shown in Fig 5C and 5D; both show an increase when the prior probability of the red edges is increased.

## Computational feasibility and timings

For an exploration of the computational feasibility and timings for carrying out analysis using our BayesNetty software, please see S2 Text.

## Results of application to an early inflammatory arthritis data set

As an illustrative example of applying BNs to a real data set, we consider our recent study [33] investigating DNA methylation as a potential mediator via which genetic variants associated with rheumatoid arthritis (RA) might confer disease risk by influencing gene expression in circulating CD4+ and B lymphocytes. In that work we applied the causal inference test (CIT) [34] to many candidate variable triplets consisting of SNP, methylation and gene expression variables, focussing on SNPs with prior evidence of association with RA. We applied BN analyses to the same data set, exploring the same candidate variable triplets as well as making use of additional individual-level data for gender, age and RA status. This provided an opportunity to compare our BN results with the original CIT analyses. We also investigated the benefit of modelling multiple variables simultaneously in one large complex network. We additionally used the final fitted network as a basis for further computer simulations investigating the performance of our imputation method compared to the usual method of discarding individuals with missing data.

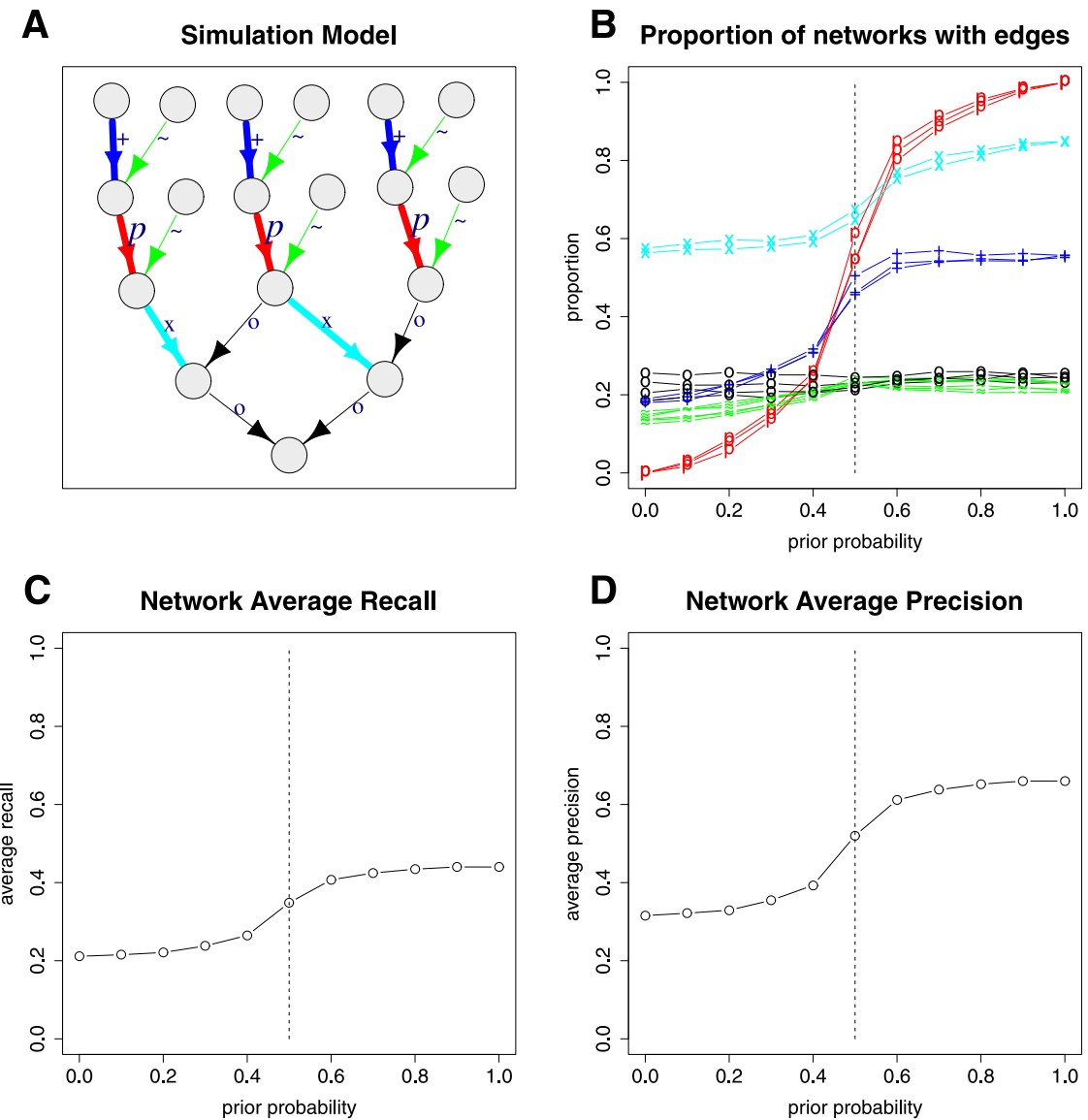

**Fig 5. Plots showing the results of 10,000 network simulations: (A) the simulation network; (B) the proportion of times each simulation edge appears in the best fit networks; (C) the average network recall; (D) the average network precision.** The effects of changing the prior probabilities of the red edges (labelled with *p*) in the plotted direction are shown in plots (B)-(D). The dashed vertical lines show when the prior probabilities are 0.5, that is, no preference in either direction. The green edges (labelled with ∼) are constrained to be in the shown direction. The colours and labels in plot (A) correspond to those in plot (B).

The early inflammatory arthritis dataset (see Methods) comprised 141 individuals with paired genotype and DNA methylation data for CD4+ T and/or B cells (from an original cohort of 280). Of these 141 individuals, only 68 had no missing data; the other 73 typically only had data for either CD4+ T cells or B cells; overall there was 16.9% missing data. The original raw data for methylation and gene expression was adjusted for batch effects and normalised as previously described [33].

**CIT and BN results.** Fig 6 plots the BN results for T cells against the CIT results, comparing the nominal CIT p-values (for an effect of methylation on gene expression) against the BN

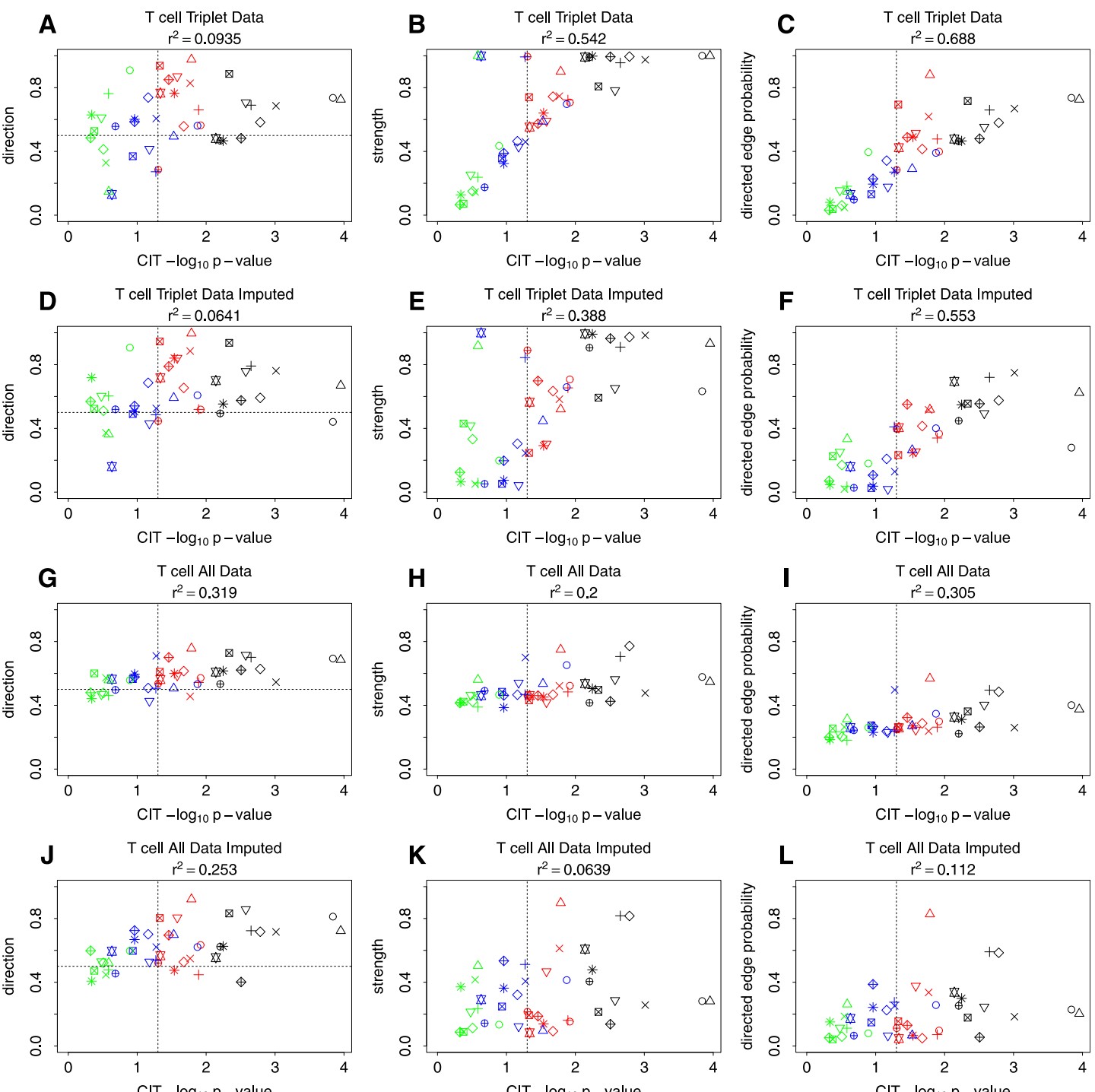

**Fig 6. T cell results.** Plots comparing the nominal −log$_{10}$ CIT p-values with BN average network posteriors of a methylation to gene expression causal effect, where each point represents a separate triplet of variables. Specific triplets are indicated by the same colour and shape in the different plots to aid in interpretation. Vertical dashed lines show a nominal p-value of 0.05. The left column shows the probability of the direction of the edge from methylation to gene expression given that it exists. A probability of less than 0.5 indicates that it is in the opposite direction. The middle column shows the strength of edges between methylation and gene expression. The right column shows the probability of the BN having a directed edge from methylation to gene expression. The top row (non-imputed data) and middle row (imputed data) show results for small BNs consisting of variable triplets plus variables for age, diagnosis and gender. The third row (non-imputed data) and fourth row (imputed data) show BN results from one large network. The squared Pearson correlation coefficient ($r^2$) is shown for each plot.

direction, strength and directed edge probability between the methylation and gene expression variables (as obtained from the average network, see Methods). The top row, panels (A), (B) and (C), shows the results for each variable triplet when analysed by discarding every individual with missing data. Panel (A) shows a weak relationship between the CIT p-values and the BN direction of the edges, although many are in the opposite direction (as the direction value is less than 0.5). The vertical line shows the nominally significant p-value threshold, 0.05. As the significance of the p-value increases the direction probability also increases. The relationship between CIT p-value and edge strength shown in Fig 6B is clearly much stronger. In Fig 6C, the directed edge probability combines the direction and strength and shows a clear relationship. As both methods use the same data, and are both effectively based on linear regression, we would expect some agreement in the results. However, the variable triplets that are considered significant by CIT do not give highly convincing causal relationships according to the BN analyses. If, for example, we were to consider that the direction value should be greater than 0.8 to be considered causal, then there would be only seven variable triplets identified (compared to 24 CIT results that are below the nominal p-value threshold). If we were to consider the overall directed edge probability, then only one variable triplet is above 0.8. This may be a reflection of limited available sample sizes for our purpose (albeit large for a study of this kind). Reassuringly, Table 1 shows that the BN results for the most significant triplets identified by Clark et al. [33] do largely corroborate the findings from the CIT (particularly when one uses a more relaxed threshold, such as BN edge probability > 0.65, and when CIT significance is assessed using a permutation-based method to calculate the FDR, which effectively provides an adjustment for multiple testing). In particular, the BN results for the three *cis*-meQTL effects that were subsequently experimentally validated by Clark et al. [33] (cg21124310/*ANKRD55*, cg07522171/*JAZF1*, and cg17134153/*FCRL3*) are amongst the most convincing.

Fig 6(D)–6(F) show a comparison of the nominal CIT p-value with the BN results obtained by applying our imputation method to each triplet separately. There are typically only a few individuals to impute for each variable triplet so the results are similar to when no imputation is used. As expected, the correlation with the CIT results is not quite as strong now that slightly different data is being used.

The plots shown in Fig 6(G)–6(I) compare the nominal CIT p-values with the BN results obtained using a large average BN fitted to all 100 variables (SNPs, T cell methylation and expression, B cell methylation and expression, gender, age and RA status) *without* using imputation. These results are again likely to be impacted by the small sample size and large number of variables; there are only 68 individuals with data for all 100 variables. This can result in a large number of false positive edges, which is reflected in panel (H) where every edge has strength of at least 0.2, as even the least significant relationships have inflated strength due to regularly appearing as (most probably false) positives. The direction is also more difficult to resolve and is shown by more methylation to gene expression edges having a direction value near to 0.5. However, there is still a weak correlation between the CIT results and the BN results. Ideally analysing all the variables together would have helped to better orientate the direction of the edges, however the resulting reduction in sample size and the large amount of variables in this particular example seem to outweigh this potential benefit.

The plots shown in Fig 6(J)–6(L) show the BN results when our imputation method is used to fit a large average BN to all 100 variables. (See S5 and S6 Figs for a visualisation of the resulting network, when the strength threshold for plotting edges is set at either 0.499 (as suggested by the formula given by Scutari and Denis [7]) or 0.6 (to achieve a sparser network)). A benefit of imputation is that it increases the sample size from 68 to 141 individuals. We would expect the correlation with CIT results to be weaker as we are effectively using different data and

**Table 1. Results for the causal inference test (CIT) in CD4+ T cells and B cells as reported in Table II of Clark et al. [33] when any nominal p-value was less than 0.05 (but focussing on those triplets also satisfying the CIT permutation false-discovery rate (FDR) threshold of 0.05, so effectively providing an adjustment for multiple testing).** When there are multiple gene expression probes for the same SNP and methylation, the most statistically significant probe was reported. Here along with the CIT p-value and CIT permutation FDR, we report the results from the BN analyses of the same data triplets with no imputation (specifically, the direction, strength and edge probabilities as calculated from the average best-fitting network).

| Gene | CpG | Lead meQTL SNP | Locus | CIT p-value | CIT Permutation FDR | BN Direction | BN Strength | BN Edge Probability |
|---|---|---|---|---|---|---|---|---|
| **CD4+ T cell** | | | | | | | | |
| ANKRD55 | cg21124310 | rs6859219 | 5q11.2 | $1.11 \times 10^{-4}$ | $7.06 \times 10^{-4}$ | 0.726 | 1 | 0.726 |
| | cg10404427 | rs6859219 | | 0.0057 | 0.0044 | 0.467 | 0.998 | 0.466 |
| | cg23343972 | rs6859219 | | 0.0062 | 0.0069 | 0.467 | 0.991 | 0.463 |
| | cg15431103 | rs6859219 | | 0.0496 | 0.0319 | 0.285 | 0.995 | 0.284 |
| JAZF1 | cg07522171 | rs2189966 | 7p15.1 | $3.97 \times 10^{-4}$ | 0.0035 | 0.69 | 0.956 | 0.66 |
| | cg11187739 | rs4722758 | | 0.0035 | 0.0044 | 0.887 | 0.808 | 0.716 |
| | cg16130019 | rs917117 | | 0.0529 | 0.0319 | 0.606 | 0.46 | 0.279 |
| ORMDL3 | cg18711369 | rs12946510 | 17q12 | $4.46 \times 10^{-4}$ | 0.0035 | 0.686 | 0.975 | 0.669 |
| | cg10909506 | rs12946510 | | 0.0016 | 0.0044 | 0.583 | 0.995 | 0.58 |
| FCRL3 | cg17134153 | rs2210913 | 1q23.1 | 0.0027 | 0.0044 | 0.707 | 0.783 | 0.554 |
| | cg01045635 | rs2210913 | | 0.012 | 0.0296 | 0.564 | 0.707 | 0.398 |
| IL6ST | cg15431103 | rs6859219 | 5q11.2 | 0.01 | 0.0296 | 0.828 | 0.747 | 0.619 |
| | cg15667493 | rs6859219 | | 0.0139 | 0.0296 | 0.558 | 0.744 | 0.415 |
| | cg10404427 | rs6859219 | | 0.0216 | 0.0305 | 0.765 | 0.641 | 0.491 |
| | cg21124310 | rs6859219 | | 0.0349 | 0.0305 | 0.85 | 0.574 | 0.488 |
| | cg23343972 | rs6859219 | | 0.0352 | 0.0305 | 0.766 | 0.553 | 0.424 |
| C11orf10 | cg16213375 | rs61897793 | 11q12.2 | 0.0163 | 0.0296 | 0.977 | 0.901 | 0.88 |
| TAX1BP1 | cg11187739 | rs4722758 | 7p15.1 | 0.047 | 0.0305 | 0.938 | 0.74 | 0.694 |
| GSDMB | cg18711369 | rs12946510 | 17q12 | 0.0277 | 0.0305 | 0.561 | 0.697 | 0.391 |
| | cg10909506 | rs12946510 | | 0.0448 | 0.0305 | 0.494 | 0.588 | 0.291 |
| **B cell** | | | | | | | | |
| FCRL3 | cg19602479 | rs2210913 | 1q23.1 | $4.69 \times 10^{-4}$ | 0.042 | 0.793 | 0.963 | 0.764 |
| | cg01045635 | rs7522061 | | $5.49 \times 10^{-4}$ | 0.042 | 0.589 | 0.98 | 0.577 |
| CCR6 | cg15222091 | rs3093025 | 6q27 | 0.0101 | 0.0966 | 0.45 | 0.824 | 0.371 |
| | cg19954286 | rs3093025 | | 0.0258 | 0.133 | 0.44 | 0.664 | 0.292 |
| | cg05094429 | rs3093025 | | 0.0347 | 0.133 | 0.351 | 0.581 | 0.204 |
| IKZF3 | cg18691862 | rs9903250 | 17q12 | 0.0249 | 0.133 | 0.732 | 0.6 | 0.439 |
| ORMDL3 | cg12749226 | rs11557466 | 17q12 | 0.0249 | 0.133 | 0.413 | 0.56 | 0.232 |

using all the variables together. The BN results do not show compelling evidence for many causal relationships between methylation and gene expression, with only three triplets having a direction value above 0.8 and only one with a directed edge probability above 0.8.

The B cell results are shown in Fig 7 and provide similar conclusions as the T cell results. The main difference is that there are more variable triplets with a causal relationship suggested in the direction of gene expression to methylation, as shown in the bottom left corner of panels (A), (D), (G) and (J).

Overall, both the T cell and B cell analyses illustrate a reasonably high correlation between the results from CIT and BN analysis when applied to exactly the same data (Figs 6C and 7C) but less correlation when our imputation approach is used to fill in the missing data, particularly when using a large average network. As this is a real data set, it is obviously not possible to say which relationships have been correctly identified, which identified relationships are false positives, and indeed which true relationships may have been missed. Further work will

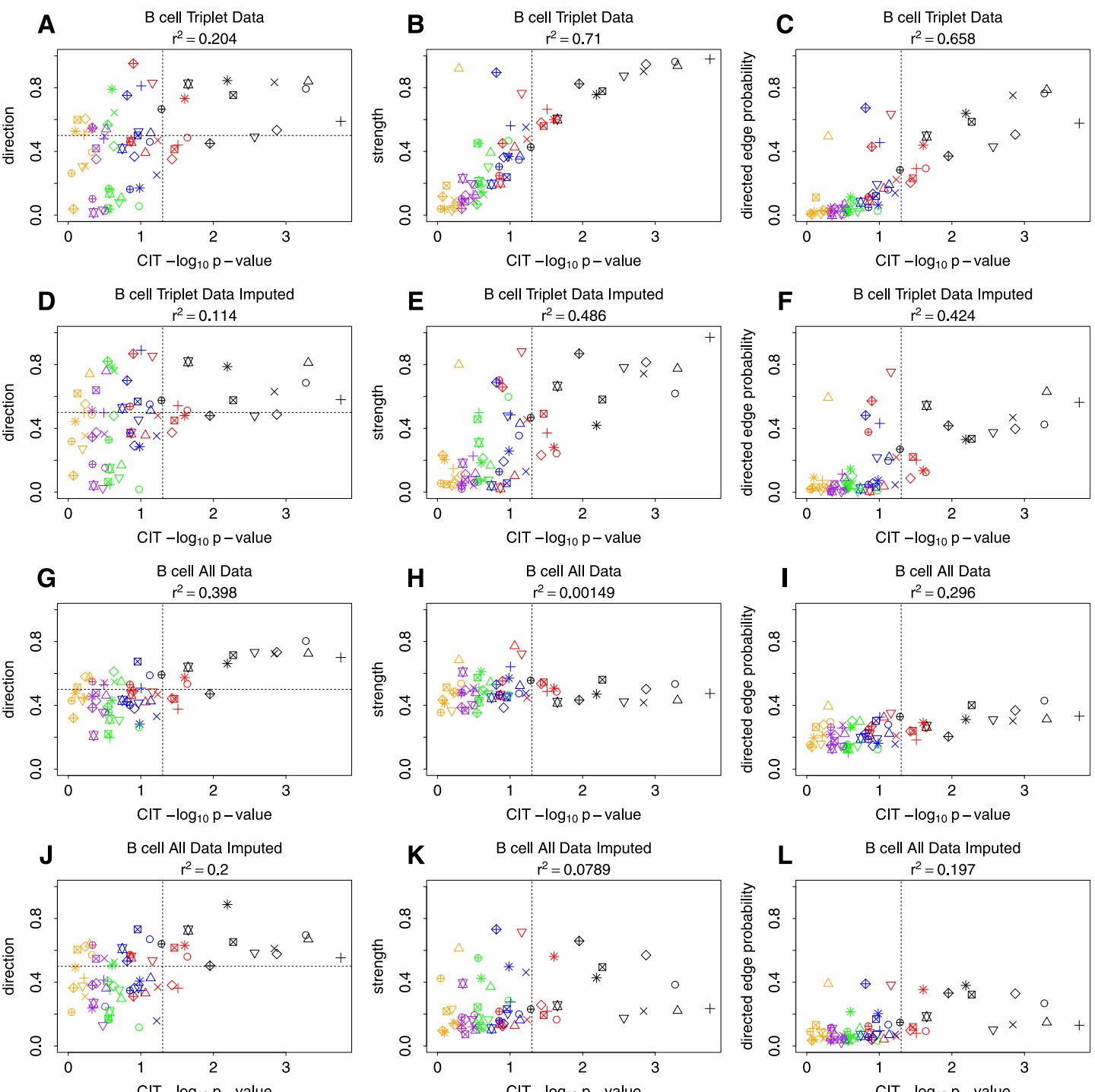

**Fig 7. B cell results.** Plots comparing the nominal $-\log_{10}$ CIT p-values with BN average network posteriors of a methylation to gene expression causal effect, where each point represents a separate triplet of variables. Specific triplets are indicated by the same colour and shape in the different plots to aid in interpretation. Vertical dashed lines show a nominal p-value of 0.05. The left column shows the probability of the direction of the edge from methylation to gene expression given that it exists. A probability of less than 0.5 indicates that it is in the opposite direction. The middle column shows the strength of edges between methylation and gene expression. The right column shows the probability of the BN having a directed edge from methylation to gene expression. The top row (non-imputed data) and middle row (imputed data) show results for small BNs consisting of variable triplets plus variables for age, diagnosis and gender. The third row (non-imputed data) and fourth row (imputed data) show BN results from one large network. The squared Pearson correlation coefficient ($r^2$) is shown for each plot.

be required to confirm (or not) the truth of the potential causal relationships identified by CIT and/or BN.

## Further computer simulations based on early inflammatory arthritis data

To provide a further comparison of the CIT and BN analysis, we simulated data to enable the plotting of receiver operating characteristic (ROC) curves. We based our simulations on the early arthritis data set [33] by first fitting a large Bayesian network to all of the methylation and gene expression data in B and T cells together with relevant SNPs, gender, age and RA status. Data were then simulated 1000 times using this best fit network as the generating model. In each simulation replicate the number of individuals and the missing data pattern in the original data was replicated, resulting in only 68 individuals with complete data from a total of 141. Triplets consisting of a SNP, methylation and gene expression that were both tested in the original study and also appeared in the best fit network were used to calculate a true positive rate. This resulted in 6 triplets for T cells and only one triplet for B cells. The false positive rate was calculated using triplets from the original study that had no direct edge between the methylation and gene expression variables in the best fit network, but where the SNP variable was causal for the methylation variable. This resulted in 10 triplets for T cells and 37 triplets for B cells. The calculated detection rates of edges were averaged over the relevant triplets and over the 1000 simulations. The p-value threshold was changed over values between 0 and 1 to calculate the true and false positive rates for CIT, whereas the probability threshold was varied between 0 and 1 for the BN analysis. The BN analyses also included variables for gender, age and RA status.

S7 Fig shows the ROC curves for the CIT and BN methods. It can be seen the BN analysis performs better in this simulation study. The results for the T cells are fairly poor for both methods which is not surprising given the small sample size. The B cell results are quite good for BN but surprisingly poor for the CIT. For any given false positive rate, the BN analysis has a higher true positive rate than the CIT.

## Discussion

BN approaches have previously been proposed as a promising tool for investigating causal relationships between biological variables, particularly when incorporating genetic variables as causal anchors [1, 2] to help resolve the directions of relationships between non-genetic variables. An appealing aspect of these approaches is their quantification of the strength of evidence provided by the data for different relationships (and indeed for the entire network) via calculation of the average network and the resulting implied strength of evidence for specific relationships. Here we show how BN analysis in data sets with mixed discrete/continuous variables can be improved by accounting for missing data. Missing data can often occur in real data sets due to cost or practical reasons. Thus, the use of imputation has great potential to boost the power of BNs to identify possible causal relationships—a key consideration when considering real-world datasets from human populations where sample sizes may be limited. Computer simulations showed that the use of our proposed imputation approaches could increase the power, in some cases dramatically, for detecting the correct causal relationships between variables. Although no method was found to be consistently optimal across all simulated scenarios, our proposed approaches generally matched or outperformed the other imputation approaches that we considered.

The first step of our default imputation algorithm (see Methods) involves taking a 90% subset of individuals and replacing any missing values in these individuals with randomly sampled values from the set of non-missing values. (See S3 Text and S8 Fig for an exploration of different

proportions rather than 90%). If there is a large amount of missing data for some variables, this random replacement (without any regard to the true structure of the relationships between variables) might be considered sub-optimal, and so we proposed an alternative "Imputed CT" method that takes the 90% subset from only the individuals that have complete data. We would anticipate this to work well in the situation where a large proportion of the individuals have complete data, but less well in the situation where there are many variables, and thus a high likelihood of every individual having one or more measurements missing. Careful examination of the missing data pattern in any given real data set may help the practitioner decide whether the default or Imputed CT algorithm would be most suitable in that particular instance. In our own computer simulations, the default and Imputed CT algorithms performed quite similarly in scenarios with reasonably large numbers of variables with relatively sparse missing data patterns that mimic the pattern we generally see in real multi-omic data sets.

Our proposed imputation approach has some superficial similarities with the multivariate imputation by chained equations (MICE) approach [31] for imputing missing data. However, from a theoretical standpoint, we do not consider our procedure to be closely related to MICE. In particular, although there is an element of randomness in our algorithm (on account of taking a 90% subset of individuals as the first step in imputing the data for an index individual), we do not perform *multiple* imputations of the missing data for any individual, i.e. in our algorithm the data are imputed only once. This contrasts with MICE in which the imputations are repeated multiple times, as the algorithm cycles over the missing data values. Within the MICE procedure, a series of regression models are run whereby each variable with missing data is modelled conditional upon the other variables in the data. This does have some similarities with our procedure, in as much as we use other "nearby" variables in the data to determine the nearest neighbour individual who should be used to impute the values of a missing variable in the index person. However, unlike MICE, we do not directly use these nearby variables to impute the value of the missing variable, but rather use them to help choose the most appropriate nearest neighbour individual.

We also considered an extension to our BN approach in the form of soft constraints, whereby a prior probability, $p$, can be assigned to any directed edge, with $1 - p$ automatically assigned to the reverse direction. This is in addition to incorporating hard constraints in the form of "white" directed edges which must be included in any BN and "black" directed edges which must be excluded. (If the white list contains both directions between two nodes, then the edge must be included but may be in any direction). The value of soft constraints is to "nudge" the edges in the right direction (if our beliefs are in fact well-founded) when there is either weak or no evidence for any direction, while allowing the data to largely overcome the specified priors if the relationships that they indicate are not, in fact, well-supported.

In our previously reported efforts to discover instances where disease-associated genetic variants might impact DNA methylation as a mechanism for altered gene expression in lymphocytes, we employed the causal inference testing (CIT) method of Millstein et al. [34] As with established BN approaches, this relies upon availability of triplets of genotype, DNA methylation and gene expression data for all individuals analysed. Although widely cited, and selected for discovery purposes in our previous study with findings subject to experimental validation, the CIT may be criticised for its susceptibility to measurement error and to the existence of unmeasured confounders such as common environmental effects [1]. It also requires the postulated causal relationships to be stated a priori (for subsequent confirmation or rejection), rather than being naturally designed for an exploratory search through the space of possible causal configurations. When applied to the same complete-data triplets, results from BN analysis and CIT were in general highly correlated. However, when BN analysis was applied using a large average network, with missing data imputed using our proposed algorithm, the

results were less concordant. In this regard our limited sample of 141 individuals, of which 73 have imputed data, may be too small to generate convincing results regarding the (possibly weak) causal relationships that may be present in a large, complex network. A role for external means to validate putative relationships identified in such settings (of the kind we have applied previously [33]) is therefore emphasised.

A limitation of BNs is the fact that analysis must be performed on individual-level data, and, unlike some competing approaches (such as MR), BN methods are not readily extended to make use of summary-level data. Whether the imputation approaches outlined here could be used as a starting point for allowing BNs to make better use of summary-level data would represent an interesting topic for future investigation.

In conclusion, we have developed an imputation approach that can be used to better identify possible causal relationships between variables such as those generated in large-scale biological experiments, leading to marked improvements in the recall and precision of directed edges in the final network of relationships identified. Our method is applicable to large, complex networks containing potentially hundreds of variables, and we provide fast, efficient, freely-available open source software implementing our proposed approach.

This research was funded in whole, or in part, by the Wellcome Trust [Grant numbers 102858/Z/13/Z and 219424/Z/19/Z]. For the purpose of open access, the author has applied a CC BY public copyright licence to any Author Accepted Manuscript version arising from this submission.

## Methods

### Ethics statement

All patient participants gave written informed consent for inclusion in the study and all associated procedures; ethical approval was obtained from the Newcastle and North Tyneside 2 Research Ethics Committee, UK (reference 12/NE/0251).

### BayesNetty software

All BN analysis in this manuscript is performed using our own software implementation, BayesNetty [2, 22]. BayesNetty is implemented in C++ using an object orientated framework for easy future development. The program and documentation with working examples are open source and freely available on the BayesNetty website [22]. The software includes the ability to plot graphs of the identified BNs using the R package igraph [35], as used for the plots throughout this manuscript.

The starting point for our method is the algorithm implemented in the R package `bnlearn` [7, 16]. The package can handle discrete, continuous and mixed discrete/continuous data. For discrete variables a multinomial distribution is assumed and for continuous variables a multivariate normal (Gaussian) distribution. In a mixed discrete/continuous network, a continuous node with discrete parents is handled by dividing the data into discrete groupings and fitting separate Gaussian distributions for each group. This can be problematic if the data set is small or some discrete categories are rare, as this may result in too little data to fit the network properly. Mixed networks are restricted such that it is not possible to have a discrete node with a continuous parent node.

Given a network structure in the form of a DAG, a network score can be calculated based on the log likelihood of the data under the assumed network structure (maximised with respect to the parameters of the assumed probability distributions). This score can be used as a measure of how well the network model describes the data, and is used to compare different networks when searching through models. In BayesNetty the network score may be set to be

based on either the log likelihood, the Akaike Information Criterion (AIC) or the Bayesian Information Criterion (BIC). We use the same definitions of scores based on the negative AIC and BIC as `bnlearn` [7], so that network scores are always negative and the larger the value, the better the fit of the network to the data.

If the number of variables (network nodes) is small, it is feasible to evaluate all possible network structures and obtain the global best fit network. Otherwise, one must search through a subset of possible network models and choose the one that fits best. BayesNetty uses a greedy algorithm for searching through network models, with the additional options of random restarts (running the algorithm a further number of times from a random starting network) and/or jitter restarts (restarting at a network given by slightly modifying the current best fit network) to avoid the algorithm sticking in a local maximum.

## Average networks

An average network, as described by Scutari and Denis [7], is a useful device to account for uncertainty in the direction of edges and in the network structure as a whole. To compute the average network, the data is bootstrapped with replacement many times (throughout this manuscript we use 1000 bootstraps) and the best fit network is fitted at each iteration. The number of times that an edge appears between two nodes in each best fit network is recorded, together with the direction. This allows us to calculate the *strength* and *direction* values (between 0 and 1) for each pair of nodes, where the strength is defined as the probability (proportion of times) that an edge appears between the two nodes and the direction is the proportion of times that the edge is in a given direction, given that it exists. If an edge has a high direction value, then it indicates that there may be a causal relationship in that direction, while, if it is near 0.5, this suggests there is little evidence provided by the data for such a relationship. The resultant average network is given by a table listing all possible edges with their strengths and directions. This typically has many unlikely edges which appeared in only a few bootstrap best fit networks and so have low strength. Thus, to plot the network while including only the most reliable edges, a strength threshold can be used to omit weak edges from the plot. Rather than choosing an ad hoc threshold, a suitable (data set dependent) choice of threshold has been proposed based upon statistical arguments [7, 36]. This choice of threshold is used as the default when plotting the average graphs calculated using BayesNetty.

## Data imputation

Here we present a new method to increase the accuracy of fitting a BN whilst accounting for missing data, where the data is in general mixed discrete/continuous. This is achieved by first imputing the missing data and then using the full (non-missing plus imputed) data to calculate a best fit or average network. The purpose of our imputation algorithm is to enable generation of the most accurate final network, which does *not* necessarily correspond to producing the most accurate estimate of the missing data per se for any particular individual or variable.

The usual approach to imputing data is by assuming a fixed BN and replacing the missing data values with their expected values. This is not possible in our context, as the "correct" BN is not known (and, in fact, it is exactly the structure of this BN that we wish to estimate). Using the (possibly small subset of) complete data to find a fixed BN with which to impute the missing values can be problematic, as this can create strong biases and artefacts in the imputed data. This then results in incorrect structural learning of the data when a final BN is fitted to the full (non-missing plus imputed) data. Therefore, our proposed imputation method aims to limit these artefacts as much as possible while also accounting for any uncertainty in the structure of the data.

We use nearest neighbour imputation, whereby missing data in one individual is replaced with data from another individual, the *nearest neighbour*. Nearest neighbour imputation is a popular method for imputing clinical data [24]. An advantage of this approach is that it does not impose any direction on the relationships between variables. Another advantage is that it can be used with mixed discrete/continuous data. The distance between individuals is calculated using a portion of the non-missing data that is assumed to be related to the missing data of interest. One challenge in nearest neighbour imputation, when there are a large number of variables, is how to select which variables to use when calculating these distances. We propose our own solution to this problem, tailored for learning BN structures, as described in the next section. To decide which variables to use for determining the nearest neighbour, we use a best fit network calculated on training data comprising a 90% subsample of the data. In S3 Text we explore the use of different fractions of the data in this step, but we still recommend using 90% as default. To avoid using the same best fit network structure to inform the imputation for every individual, we resample a new set of training data before imputing the missing data for each individual that has missing data. This provides a way to incorporate model uncertainty in the best fit network that is used to inform the imputation.

**Data imputation algorithm.** We assume that the missing data is missing at random (MAR), that is, the missingness of a given variable is dependent only on the values of other measured variables, with no biases in relation to the values that are missing (for example, higher values are more likely to be missing). Missing data is imputed for each target "index" individual separately using the following steps:

1. A 90% subset of individuals chosen from all the individuals (thus potentially including the index case) is taken without replacement. The missing data values for each variable within this subset are replaced with randomly sampled values (with replacement) from the set of non-missing values.

2. A best fit BN is found using this data set. Any algorithm to find the best fit network can be used and in this manuscript we use the greedy search algorithm with a Bayesian information criterion (BIC) network score.

3. The missing data is imputed using our own version of nearest neighbour imputation with the variables used to define the nearest neighbour selected on the basis of the best fit network. For each variable with missing data, a list of all the other variables that have connecting edges and have non-missing data for the index individual is constructed. These variables are known as the *nearby variables*. These nearby variables are then used to calculate the distance between the index individual and every other individual that has non-missing data for both these nearby variables and for the variable with missing data that needs to be imputed. The individual with the smallest distance is designated as the nearest neighbour; when there are multiple individuals with the same smallest distance one is randomly chosen. The overall distance is calculated as the sum of distances from every nearby variable. For continuous variables that distance is defined by the difference squared divided by the variance of the variable, that is, the normalised difference. The variance is estimated from the complete data for that variable. For discrete variables the difference is defined by 1 if the categories are different and 0 if they are the same. If an imputed variable has no nearby variables, then a random individual (with non-missing data for this variable) is chosen as the nearest neighbour.

4. The missing data for the relevant variable in the index case is replaced by the data for that variable from the nearest neighbour.

**Modifications to the data imputation algorithm.** As well as using a network to select the nearby variables, another difference between our method and other nearest neighbour implementations relates to a modification not yet mentioned: in some instances a new variable which is not in the original data is constructed and used as a nearby variable. Suppose a continuous variable $z$ has missing data and is a parent of continuous variable $y$ which also has another continuous parent variable $x$. A new nearby variable, $v$, is defined by $v = y - \beta_x x = \beta_0 + \beta_z z$ where the $\beta$s are the regression coefficients given by $y = \beta_0 + \beta_x x + \beta_z z + \epsilon$, when $y$ is regressed on $x$ and $z$ using standard linear regression. We then use variable $v$ rather than $y$ as a nearby variable for informing the choice of nearest neighbour to impute the missing value of $z$, since $y$ depends on both $x$ and $z$, whereas $v$ depends mostly on $z$. The definition of the new nearby variable is extended naturally if there are multiple continuous variables that are parents of $y$. If any of the continuous parent variables of $y$ have missing data for the individual that is being imputed, then this variable is simply not used in the newly defined variable $v$, and so if all continuous parent variables have missing data this means that $y$ is used as the nearby variable. If $z$ is discrete, then it is imputed in the usual way, as it is not possible to perform the necessary linear regression as the correct discrete partition of the data is unknown.

We also define an alternative version of our imputation method by making a change to the first step. Specifically we take a 90% subset from the individuals that have complete data (and we thus do not need to randomly replace any missing values). We refer to this alternative method as "*imputation with complete training data*" and denote it by "Imputed CT" in figures. The main disadvantage of this alternative method is that it requires sufficient individuals with complete data, and thus the training data set can be much smaller, resulting in less accurate imputation. The advantage comes when there is a large amount of missing data for some variables, and so we avoid weakening the strength of the edges connected to these variables in the training data set, assuming enough complete data is available.

## Computer simulations to evaluate performance

We carried out a number of different simulation studies to evaluate the performance of our proposed approaches. For more details of these, please see S1 Text.

## Soft constraints

There may be prior belief that the direction of causality between two variables is in a certain direction. To take account of this, we can define a weighted network score based on the Bayesian Information Criterion (BIC) score where each directed edge, $e$, between two specified nodes is given a prior probability, $p(e)$, with the reverse direction automatically set to $1 - p(e)$. Define $x$ to be the observed data, $M$ the network and $\hat{L}$ the maximised likelihood. The sample size of the data set is given by $n$ and $k$ is the number of edges. From the derivation of the definition of the BIC we have:

$$-2 \ln p(x|M) \approx BIC = -2 \ln \hat{L} + k(\ln n - \ln 2\pi)$$

$$\ln p(x|M) \approx -\frac{1}{2} BIC = \ln \hat{L} - \frac{k}{2}(\ln n - \ln 2\pi)$$

$$p(x|M) \approx \frac{\hat{L}}{\left(\frac{n}{2\pi}\right)^{\frac{k}{2}}} \propto \frac{\hat{L}}{n^{\frac{k}{2}}}$$

Therefore, if we multiply the contribution that each edge, $e$, makes to the score by the prior

probability that an edge exists in this direction, we have:

$$p(x|M) \times \prod_e p(e) \propto \frac{\hat{L}}{n^{\frac{k}{2}}} \times \prod_e p(e)$$

$$\ln p(x|M) + \sum_e p(e) = \ln \hat{L} - \frac{k}{2} \ln n + \sum_e \ln p(e) + constant$$

And so, using the usual definition of BIC as $\ln \hat{L} - \frac{k}{2} \ln n$, we define our new weighted score as:

$$\text{Weighted Score} = BIC + \sum_e \ln p(e)$$

An edge with the prior probability of 0.5 will have no preference in either direction and the best direction will be given by the usual BIC score, whereas an edge with a prior direction probability of 0 will result in a score of minus infinity and the network being rejected. A score strictly between 0.5 and 1 (not including 0.5 and 1) will favour an edge in a certain direction but will allow it to be in the other direction if the data are sufficient to suggest otherwise.

## Data application: Early inflammatory arthritis and intermediate biological data

As an illustrative example of applying BN analysis to a real data set, we used data from a recent study [33] of RA and the possible influences of methylation on gene expression in CD4+ T cells and B cells.

**Early inflammatory arthritis data.**   The RA data set comprised 141 individuals, of whom only 68 had no missing data; the other 73 individuals typically only had data for either CD4+ T cells or B cells. Raw data for methylation and gene expression was adjusted for batch effects and normalised as described [33].

**Variable triplets.**   For CD4+ T cells there were 42 variable triplets consisting of a SNP, methylation of a CpG and a gene expression probe that were identified in the original study [33] as being of interest based on instances in which CpG methylation was simultaneously correlated with risk variants (meQTLs) and transcript levels of genes in cis (eQTMs). For B cells there were 65 such variable triplets. We applied BN analyses to each candidate variable triplet along with the additional variables coding for gender, age and RA status. For each triplet we applied BN analyses without imputation, so that only 68 individuals were used. We also analysed the data using our imputation approach. For each triplet we removed every individual who had neither methylation nor gene expression data, so that for the T cell data there were typically around 103 individuals in total with only three individuals having missing data, while for the B cells there were typically 113 individuals in total with only 10 individuals having missing data. We calculated an average BN giving estimates of the direction of causality between methylation and gene expression variables.

**Use of a large BN to model the candidate triplets.**   We also applied BN analysis using every variable from the candidate variable triplets, together with gender, age and RA diagnosis, which gave a total of 100 variables. We calculated an average BN with and without imputation. The resultant average BNs were then used to estimate any causal relationships between the candidate methylation and gene expression variables. In theory, the inclusion of all variables simultaneously could help to better orientate the direction of causality of any relationships.

### Quantification of data underlying plots and graphs

We quantified the data underlying the various plots and graphs resulting from our analyses in S1 Spreadsheet.

### Supporting information

**S1 Fig. Line graphs showing the recall and precision.** Shown are results for the best fit network found by BayesNetty (and alternative approaches) when the simulation network was $A \rightarrow B \rightarrow C \rightarrow D \rightarrow E$ for different data sets. The asterisks denote which variables had missing values for 1800 individuals from a total of 2000 individuals. The parameter $\beta$ denotes the model strength that was used to simulate the data. Full: the data consisted of the original simulation data with no missing values. Imputed: the data was imputed using our default algorithm. Reduced: the data was reduced to the 200 individuals with no missing values. Imputed CT: the data was imputed with complete data training. EM: the data was imputed with an expectation-maximisation algorithm in R. Random: missing data was replaced by random values drawn from the same variable. Mean: missing data was replaced by the mean value of the same variable. All NN: the data was imputed using our default algorithm but with all variables used to inform the choice of nearest neighbour. MICE: the data was imputed using the multivariate imputation by chained equations approach.
(EPS)

**S2 Fig. Line graphs showing the recall and precision.** Shown are results for the best fit network found by BayesNetty (and alternative approaches) when the simulation network was given by three different networks in the `bnlearn` Bayesian Network Repository. Simulated data consisted of 500 individuals of which 450 had each variable set to missing with various probabilities as indicated on the horizontal axis. Full: The green line shows when the data consisted of the original simulation data with no missing values, so the missing percentage does not apply to this line. Imputed: the data was imputed using our default algorithm. Reduced: the data was reduced to around 50 individuals with no missing values. Imputed CT: the data was imputed with complete data training. EM: the data was imputed with an expectation-maximisation algorithm in R. Random: missing data was replaced by random values drawn from the same variable. Mean: missing data was replaced by the mean value of the same variable. All NN: the data was imputed using our default algorithm but with all variables used to inform the choice of nearest neighbour. MICE: the data was imputed using the multivariate imputation by chained equations approach.
(EPS)

**S3 Fig. Plots showing the proportion of times the best fit BN has a directed edge from *X* to *Y*.** Shown are results from 1000 data simulations when the prior probability of a directed edge from *X* to *Y* is varied. (A) The plot on the left shows a model with an effect from *A* to *X* to *Y*. (B) The plot on the right an effect from *A* and *Y* to *X*. The effect size between *X* and *Y* is fixed at 0.5. The different colour lines show how the results differ when the effect size is varied between *A* and *X* (where a value of 0 gives no effect).
(EPS)

**S4 Fig. Plots showing the proportion of times the best fit BN has a directed edge from *B* to *X* (left column) or from *X* to *Y* (right column) for 1000 data simulations.** The simulation model has effects from *A* and *B* to *X* and from *X* to *Y*. The effect sizes from *B* to *X* and from *X* to *Y* are fixed at 0.3. The effect size from *A* to *X*, *a*, is varied at values 0.1, 0.3 and 0.5 and shown in different plots. The result of changing the prior probabilities of directed edges from *B* to *X* and *X* to *Y* is shown in the plots. The white dashed lines show when the prior

probabilities are 0.5, that is, no preference in either direction. In panel (A) we are detecting the edge from $X$ to $Y$ for the weakest effect of edge $A$ to $X$. Thus as the prior probability from $X$ to $Y$ increases so does the proportion of times this edge is detected in the best fit BN. When the prior probability of $B$ to $X$ increases we interestingly see that the proportion of times $X$ to $Y$ is detected also increases even though it is a different edge. The edge from $B$ to $X$ helps to better orientate the edge from $X$ to $Y$, therefore when it is present the edge from $X$ to $Y$ is also more likely to be present. When the prior probability for the edge $B$ to $X$ is in the opposite direction this support is missing and so the proportion of times $X$ to $Y$ is detected decreases slightly, although it does not approach 0 as there is still some evidence for an edge from $X$ to $Y$. When the strength of the edge from $A$ to $X$ is increased as shown in panels (C) and (E), the edge from $B$ to $X$ becomes less relevant as the edge from $A$ to $X$ already provides a lot of support. The proportion of times the edges $X$ to $Y$ is detected is then largely determined by the prior probability of this edge. In panel (B) the proportion of times an edge is detected from $B$ to $X$ is shown for the weakest setting of the edge from $A$ to $X$. As expected when the prior probability of the edge from $B$ to $X$ is increased the proportion of times the edge is detected also increases. Similar to before it is interesting to observe that as the prior probability from $X$ to $Y$ increases the proportion of times an edge from $B$ to $X$ is detected also increases. This shows that when edges are correctly orientated they can help better orientate other edges in the network. As the strength of the edge from $A$ to $X$ increases, shown in panels (D) and (F), the impact of the prior probability of $X$ to $Y$ is reduced as the edge $A$ to $X$ provides some support in orientating both $X$ to $Y$ and $B$ to $X$. Only when the prior probability of $X$ to $Y$ is set strongly in the wrong direction do we see a large impact on the proportion of times $B$ to $X$ is detected.
(EPS)

**S5 Fig. Average BN using imputed data of all variables from candidate variable triplets together with variables for gender, age and RA diagnosis.** A strength threshold (0.499) was applied as suggested by the formula given by Scutari and Denis [7]. Edges are labelled with the probability that they exist (strength), and in brackets the probability that they exist in the shown direction (given that they exist). The thickness of the edges is proportional to the edge strength. The nodes are such that: Dark blue (with gene name prefix) are B cell gene expression; dark pink (with gene name prefix) are T cell gene expression; light blue (with "b_" prefix) are B cell methylation; light pink (and "t_" prefix) are T cell methylation; green are SNPs; red is RA diagnosis; beige is age; and yellow is gender. With the large number of variables and low number of individuals, it is unfortunately probably not very accurate as the section of the manuscript estimating the recall and precision shows. It can be seen that variables of the same kind are generally connected to one another. For example, the gene expression measurements that are prefixed with the same gene name tend to be connected, such as the FCRL3 B cell expression measures which are strongly connected together. The direction between these variables are all around 0.5 suggesting they are not causal on one another.
(EPS)

**S6 Fig. Average BN using imputed data of all variables from candidate variable triplets together with variables for gender, age and RA diagnosis.** A strong strength threshold (0.6) was applied to reduce the number of edges shown. Edges are labelled with the probability that they exist (strength), and in brackets the probability that they exist in the shown direction (given that they exist). The thickness of the edges is proportional to the edge strength. The nodes are such that: Dark blue (with gene name prefix) are B cell gene expression; dark pink (with gene name prefix) are T cell gene expression; light blue (with "b_" prefix) are B cell methylation; light pink (and "t_" prefix) are T cell methylation; green are SNPs; red is RA diagnosis; beige is age; and yellow is gender. There are less edges between different types of

variables in this network plot compared to S5 Fig, suggesting that these edges are less likely to be false positive edges.
(EPS)

**S7 Fig. ROC curves for detecting a causal relationship from methylation to gene expression.** CIT denotes the casual inference test and BN denotes Bayesian network analysis. Positive and negative detection rates were calculated using averages taken from simulated data with 1000 replications.
(EPS)

**S8 Fig. Plots showing the average recall and precision using random training data (RT) and complete training data (CT) using 1000 simulations for each data point.** Each line shows the percentage of missing data in 450 individuals out of the 500 individuals.
(EPS)

**S1 Spreadsheet. Quantification of data shown in the figures and supplementary figures.** (XLSX)

**S1 Text. Computer simulations to evaluate performance.**
(PDF)

**S2 Text. Computational considerations and timings of BayesNetty, including use of the Open Message Passing Interface (MPI) for parallel processing.**
(PDF)

**S3 Text. Exploration of the effect of varying the subset percentage size taken in step 1 of the BayesNetty imputation algorithm.**
(PDF)

## Acknowledgments

This research benefited from infrastructural support (IS-BRC-1215-20001) by the National Institute for Health Research Newcastle Biomedical Research Centre (https://www. newcastlebrc.nihr.ac.uk/) based at Newcastle Hospitals NHS Foundation Trust and Newcastle University. The views expressed are those of the author(s) and not necessarily those of the NHS, the NIHR or the Department of Health.

## Author Contributions

**Conceptualization:** Louise N. Reynard, Arthur G. Pratt, Heather J. Cordell.

**Formal analysis:** Richard Howey, Alexander D. Clark, Najib Naamane.

**Funding acquisition:** Arthur G. Pratt, Heather J. Cordell.

**Investigation:** Richard Howey, Alexander D. Clark, Najib Naamane.

**Methodology:** Richard Howey, Heather J. Cordell.

**Software:** Richard Howey.

**Supervision:** Louise N. Reynard, Arthur G. Pratt, Heather J. Cordell.

**Writing – original draft:** Richard Howey, Heather J. Cordell.

**Writing – review & editing:** Alexander D. Clark, Najib Naamane, Louise N. Reynard, Arthur G. Pratt.

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
