## [Decision Letter · Decision Letter 0]

16 Mar 2021

Dear Heather,

Thank you very much for submitting your Research Article entitled 'A Bayesian network approach incorporating imputation of missing data enables exploratory analysis of complex causal biological relationships' to PLOS Genetics.

You can find below the comments from three reviewers, recommending major revisions. While the paper has been well received, all reviewers agree that the methodological contributions should be supported by more comprehensive literature review and by a more detailed discussion of the underlying assumptions. They also ask for some clarifications on the experimental evaluation.

If you decide to revise the manuscript for further consideration at PLOS Genetics, please aim to resubmit within the next 60 days, unless it will take extra time to address the concerns of the reviewers, in which case we would appreciate an expected resubmission date by email to plosgenetics@plos.org.

[LINK]

We are sorry that we cannot be more positive about your manuscript at this stage. Please do not hesitate to contact us if you have any concerns or questions.

Yours sincerely,

Marco Scutari

Guest Editor

PLOS Genetics

David Balding

Section Editor: Methods

PLOS Genetics

Reviewer's Responses to Questions

**Comments to the Authors:**

Reviewer #1: This is pretty good stuff. It’s very thorough. Very well-written

I have a couple major points:

Causality (or lack thereof in BN semantics): The authors should address the fact that edges in BNs aren’t causal and only represent statistical dependence (or, rather, conditional independence) relationships. There’s no expectation that the directions of arrows in a BN really mean that much, since, under many conditions, there’s no statistical difference. Consider the paragraph in lines 12-21. Nothing about BN semantics implies causality. The authors should either talk about under which conditions causality can be inferred (such as in Heckerman, https://arxiv.org/abs/1302.4958 ), or drop the references to causality (Or cite some of the causal-BN proponents in a balanced discussion). The authors should also consider if edges they’re trying to recreate in networks are edges that are constrained to point in one direction or if they’re edges that could have been in the other direction. This has implications for how their edge-recreation algorithms are scored.

Why throw out whole samples?: I’m not sure, but I assume that when learning BNs there’s no need to throw out a sample simply because it has a few missing values. Since BNs are built using local conditional probability tables, each local dependency computation can be computed with as much data as is available for the variables in question. So a sample can easily contribute to the CPTs for which it contains values, while being skipped for the computations involving the values it is missing. It would seem this is the natural baseline for comparison of imputation methods.

There’s a lot of data double dipping, and I suspect this leads to inflated results. The whole scheme for estimating precision and recall seems to have been created de novo, since no citations are provided. In the description of this method, the same imputation method is used in step 1 and in step 5(ii). Since the methods tested in step 5 are compared to the results of step 1, one assumes that method (ii) should do the best by this metric since it’s the same method that was used to generate the data. This is what is played out in the results.

A philosophical point:

The problem of recapitulating edges from biological networks is very difficult. It’s also difficult to know if it’s being done properly, since it’s hard to find such networks with large datasets where the true network is known with certainty. Thus, authors usually present toy examples of networks with a handful of nodes and simulated data, then argue that their recapitulation algorithm works on this toy example, so it might therefore work on real data. However, the gulf between a toy network of 5 nodes and actual biology on an omics scale is staggering. It might be interesting to see some kind of samplesize calculation that estimated the number of samples required to do this on a much bigger scale.

Minor Points:

Introduction:

There’s some confusion at the beginning about whether the point of this is to recapitulate known network’s structure or to use the BN to perform imputation of missing data. Bit of Chicken and Egg issue, really. It’s not clear which of these is the primary task.

Methods:

Couldn’t imputation be improved by not throwing out “nearby variables” that have 1 or more missing value?

Not sure if they justify the substitution of nearest neighbor variables for the case of continuous variables. I guess the idea is to use residuals to get a combination of near variables, but it’s not clear why this is preferable. Does just using the near variables result in poor performance?

Interesting to include variable Bayesian priors on network edges based on belief, however I think the problem with this is finding a principled way of assigning priors without biasing your results. If the authors have intuitions about how best to do so, it would be a welcome contribution.

The authors are talking about methylation of SNPs, and then say that “100” is a “large BN”. Shouldn’t there be about 400k methylation sites and 6 million SNPs?

Results:

There’s no method explanation for how the power plots of Figure 1 were generated. Was it using the Precision and Recall methodology?

Figure 2 seems to be based on the Precision Recall methodology that biases toward the authors’ imputation method.

The CIT and BN results might benefit from an AUC analysis where the accuracy across various thresholds can be visualized. This may be aide the discussion about thresholds of 0.8 and of 0.65.

Comparison of CIT and BN results are also a little unsatisfying. As I understand it, the CIT results are from the authors’ prior work, and there’s no expectation that they’re more correct than the BN results. This portion is sort of interesting, and it would probably be remiss of the authors not to make such a comparison if they’re comparing two methods for doing the same thing on the same data; but it could perhaps be summarized or moved to the supplement for the reader interested in comparing these methods.

Reviewer #2: Review

In this paper the authors present a novel method to account for missing data when building a Bayesian network based on high dimensional genetic data. In this setting, restricting the analysis to individuals with complete data (i.e. no missing values) can result in a dramatic loss of precision and potentially bias. The author’s propose the use of an algorithm that incorporates nearest neighbour imputation to fill in the missing data.

Simulations are performed to assess the performance of the approach, software is provided and the method is nicely demonstrated on an applied example

Comments

Page 8:

Line 67 Some use the term `complete case’ in place of `Full’ which could be mentioned

Line 82: Can you make clear here or earlier on that you only consider DAGs (if that’s the case?)

Line 94: Can you clarify if the ground truth of the 46, 37 and 27-node network data sets in the BNLearn repository is known?

Page 9,

Line 120

Presumably priors must always be specified, they are not an optional extra? I think you may discuss this later on in the methods, but I was a bit confused at this point. Can you say something here about how one should incorporate prior information, or set priors if they want them to be informative/non-informative?

Line 123: Is A an instrument for X?

Line 127

Can you say anything general about when increasing the prior for an edge will effect the chances of correctly/incorrectly detecting an edge between any other two variables? Is it as simple as saying that if there is a path from , say, A to Y in the DAG ? i.e. The pathway is `open’

Line 136. You mention that some edges are not assigned prior probabilities at all. Now I’m confused! I would have thought that you would have to assign a prior for a Bayesian method to ?

Page 10

Line 179 Did you use a multiplicity correction in your work?

Page 14 lines 297-8 I’m slightly confused by the statement `applying a prior probability conditional on the event that it exists’? Can you clarify, this (see earlier comments) and that you are using soft and hard priors

Page 15

Line 377. Is this threshold 0.6?

Page 16

Line 408 Not sure if this definition of MAR is correct. Yes outcome dependent missingness is MNAR. But it is possible that missing values are higher than average (or biased), say, but are correctly imputed using nearest neighbour methods (that is, using observed data)

Line 436. Is this approach related to the MICE procedure (multiple imputation using chained equations)?

Reviewer #3: This manuscript describes an adaptation of an existing algorithm for fitting Bayesian networks (BN) when some data are missing in a subset of samples, by using nearest neighbour imputation, where the variables to be used in learning the nearest neighbour are themselves learnt from a BN.

This is a nice idea, and the approach is evaluated through simulation and real data analysis. However, it is not clear which of the two imputation methods are optimal in which situations, when they outperform other baselines, computational requirements, and whether the proposed method actually outperforms CIT or not in the real data analysis - how can this comparison be evaluated without any ground truth knowledge?

Finally, the manuscript is in need of some rewrites, to better cover the area of Bayesian networks generally and BayesNetty specifically in the introduction, and to conform to a manuscript structure that puts Methods at the end.

* Major comments *

Fig 1 caption: “The title of each plot shows the simulating model and the asterisk denotes which variable or variables are missing in 1800 missing individuals from a total of 2000 individuals (only one variable is missing for each individual).”

Comment: This is quite a significant amount of missingness (90%) and a large number of samples compared to the real data (141 individuals, roughly 25-50% missingness based on the description in lines 168-169) and to the Bayesian Network Repository datasets (500 individuals, approximately 10-40% missing). The large sample size in this 3 variable simulation might help explain why the Reduced method seems to do so well here. Including a simulation study with numbers more similar to that of the real dataset would be an improvement. If not I think justification is needed of why larger sample numbers are used for the small graph simulations.

Lines 224-228: “Overall, both the T cell and B cell analyses illustrate a reasonably high correlation between the results from CIT and BN analysis when applied to exactly the same data (Fig 6 (c) and Fig 7 (c)) but less correlation when our imputation approach is used to fill in the missing data, particularly when using a large average network. Further work will be required to confirm (or not) the truth of the causal relationships identified by CIT.”

Comment: It’s unclear what value this comparison (of the results from CIT and from BN analysis) adds.

The layout of the paper is difficult, with results on the performance of the method preceding description of the method itself. Some understanding of the method is needed by the reader to have some intuition when reviewing these results. A first section of the results giving an overview of the method would be useful.

I also think a more complete review of imputation in Bayesian networks would be useful. For example, Bayesian networks themselves have been used often as a method for imputation, and different imputation methods when fitting Bayesian methods have been considered (eg BCPA, LLS). BayesNetty is not described in any detail so that I can understand what it does, how it does that, which is important as this imputation is built upon it. Why use BayesNetty to build upon, other than because it is the authors’ earlier work?

There is limited information on running time. How long does each run with the real data example take? How does running time scale with the number of individuals/variables? The imputation step alone, which must be run once per individual with missing data, seems intensive.

Two imputation approaches are proposed, which seem to beat each other in different circumstances. Could the authors give some intuition about why this is, and which method is more appropriate for different datasets? Further, there is no exploration about the optimal parameters for imputation. In the either imputation method, should the fraction of the data used for generating the nearest neighbours vary with sample size? What is the advantage of using 90% of the data each time, rather than all the data? Would replacing missing values with the mean for each variable rather than a random value change things?

The method isn’t universally optimal, particularly with small datasets: e.g. in Figure 1 the authors show that imputation is comparable with EM, and is sometimes worse than using only 10% of the dataset, but that is for just 3 variables. In Fig 4 and Fig 8a-d, they show that imputation is comparable with random initialisation of variables. In Fig 8e-f the imputation methods do outperform random and reduced, though it’s a shame not to see EM in this figure. They claim their method is better on larger networks, so possibly they should include more datasets where their method excels (or e.g. move fig s2 to main, since it actually shows their method narrowly beating EM and random).

It’s unclear how the soft constraints section fits into the rest of the paper, which focuses on imputation. The largest dataset used for analysis of soft constraints had only ~20 nodes, which is small compared to many simulations used in the paper. What link does the soft constraints method have to the imputation method?

It’s difficult to understand what is learned by comparing the results on the real datasets obtained by this method relative to an existing method, especially when the outputs of the methods are different (Bayesian estimates vs frequentist p-values). The output of the existing method (CIT) cannot be treated as a ground truth.

* Minor comments *

How many repeats were run for each simulation?

Line 61 - “Fig 1 shows the power (or Type I error when β=0)”

Comment: I’m unclear on the need or relevance of a hypothesis testing framework here, and am concerned that, as it stands, the bracketed clause reads as an (incorrect) redefinition of the statistical term power. Either the null and alternative hypotheses should be clearly stated, the authors should use the same recall/precision used for future simulation models, or use e.g. “Proportion of runs where best fit network matches”.

Line 121 - “We start with a simple model to demonstrate the effect of changing the prior probability of an edge from X to Y”

Comment: Add clarification that the prior is conditional on the edge existing i.e. it is a prior on direction. E.g. “A prior probability is assigned to the direction of an edge between X and Y, conditional on the event that the edge exists.”

Line 125 - “Provided βA is non-zero, increasing the prior probability of an edge from X to Y increases the power for its detection when it exists (S3 Fig (a)), while only increasing the chance of a false positive detection (S3 Fig (b)) when the directed edge is from Y to X.”

Comment: I found this difficult to understand, particularly from “only increasing the chance…” onwards. Fig S3a seems to have points in the bottom left quadrant, corresponding to when the prior favours Y->X and the best fit networks favour Y->X, which are false positives as in Fig S3a the true model has X->Y. In Fig S3b, there are points in the top right quadrant, corresponding to when the prior favours X->Y and the best fit networks favour X->Y, which again are false positives. So it seems to show that, whether the edge is really X->Y or Y->X there are false positives, though there are few false positives for βA >= 0.4.

Line 169 - “Of these 141 individuals, only 68 had no missing data; the other 73 typically only had data for either CD4+ T cells or B cells.”

Comment: Could the authors add information about percentage of values missing, to allow easy comparison with the simulated datasets used?

Line 445 - “We also define an alternative version of our imputation method by making a change to the first step. Specifically we take a 90% subset from the individuals that have complete data (and we thus do not need to randomly replace any missing values). We refer to this alternative method as “imputation with complete training data” and denote it by “imputed CT” in figures.”

Comment: This method is not consistently included in the analysis and not discussed in the discussion.

Fig 3

Comment: The grey lines (edges missing from the graph), as the authors note, are “almost invisible”. Perhaps the grey lines could be made darker, and the orange lines more distinct from the red. A legend would be very helpful. Could a network for Random be included?

Fig 4

Comment: Is there a reason for Imputed-CT and EM to be excluded from this graph? In general it would be helpful for comparison if all methods were included in all plots (e.g. EM is also excluded from Fig 8)

Figs 6, 7

Comment: What do the different colours and shapes for the points mean? Are they necessary?

Text S2: Computational feasibility and timings.

Comment: It would be useful to be able to compare timings between the methods used in the paper i.e. Imputed, Imputed-CT, Reduced, Full and EM. It’s useful to be able to see how the timings scale with the number of variables (nodes), but it would also be useful to see how it scales with the number of samples. In the tables in S2, I would suggest changing the label of “Simple Parallel” to “Parallel (1000)” to make it clear it requires 1000 parallel processes.

**Have all data underlying the figures and results presented in the manuscript been provided?**

Reviewer #1: Yes

Reviewer #2: Yes

Reviewer #3: **No: **Could not find scripts and data used for simulations. Not clear which scripts in https://github.com/aclark5/Lymphocyte_meQTL relate to this paper. Annotation could be improved (eg - give a hint in the README which scripts to look at!)

PLOS authors have the option to publish the peer review history of their article (what does this mean?). If published, this will include your full peer review and any attached files.

Reviewer #1: No

Reviewer #2: **Yes: **Jack Bowden

Reviewer #3: No

---

## [Decision Letter · Decision Letter 1]

2 Jul 2021

Dear Dr Cordell,

Thank you very much for submitting your Research Article entitled 'A Bayesian network approach incorporating imputation of missing data enables exploratory analysis of complex causal biological relationships' to PLOS Genetics.

The manuscript was fully evaluated at the editorial level and by independent peer reviewers. The reviewers appreciated the attention to an important topic but identified some concerns that we ask you address in a revised manuscript.

We find that the authors have addressed most of the comments that the reviewers made on the previous submission. However, Reviewer #1 feels that some of the points he made in his review have not been addressed in full. In particular, we agree that the paper could be made more compact by dropping material that is less central; and that any apparent issues with data double-dipping should be clarified. Please also consider at least one other method of imputation to compare with the proposed method.

We therefore ask you to modify the manuscript according to the review recommendations. Your revisions should address the specific points made by each reviewer.

[LINK]

Yours sincerely,

Marco Scutari

Guest Editor

PLOS Genetics

David Balding

Section Editor: Methods

PLOS Genetics

Reviewer's Responses to Questions

**Comments to the Authors:**

Reviewer #1: Well, I feel I understand this paper better now, although certainly it is not a complete understanding. Much of the Bayesian network discussion and material is actually tangential to the main purpose. As I understand it, the authors are presenting a new method of imputation specifically just for BN edge reconstruction (and specifically just for use with their own BN method, BayesNetty), which is a combination of preliminary BN structure learning and nearest-neighbor imputation, and therefore they only test their imputation by the quality of BN edge reconstruction the imputation enables.

First, if they’re going to propose new methods of imputation, they should be tested against existing methods of imputation. This is mostly not done. There’s no testing against imputing by the mean, hotdecking, regression based, existing nearest-neighbor methods, or other probabilistic imputation, except for EM, which they then fail to use in many of the cases considered (they claim that EM is unsuitable to mixed networks, but the basic idea of EM is very simple and very general, I find this argument lacking merit. However, even a straightforward extension of EM to the hybrid continuous-discrete network case would be more appropriate to a separate paper).

Second, if they’re going to propose methods of imputation for edge reconstruction, they should test imputation with multiple methods of BN edge reconstruction, instead of just their own software BayesNetty. It seems like hillclimbing, K2, basically any algorithm, with or without backtracking and restarts, could be used for either the BN-aided imputation portion of the algorithm, or for the BN edge reconstruction portion of this endeavor that serves as the metric.

Third, the idea that the best metric of imputation-quality is BN edge reconstruction requires some justification. What is the difference between using this metric and the metric of just “accuracy of imputation”? A comparison could easily be presented using the existing simulation studies.

If they could fix the first problem, this work would be of interest to a limited group: people using BayesNetty for edge reconstruction in datasets with missing data, who decide that imputation is their best option. A re-titling would help make the work more clear:

“BN-assisted nearest-neighbor imputation of missing data to optimize edge reconstruction of BayesNetty software”

Length / Complexity / Focus:

The paper is now over 30 pages of single-spaced text with another 8 pages of figures, and 12 separate supplemental materials. This should probably be split into two papers (or more), or greatly streamlined.

At a first glance, I would suggest cutting: (1) everything related to “soft constraints”. (2) all the CIT vs BayesNetty comparisons. (3) every mention of “causality” and related text (see below).

Minor new points:

I suggest they provide a name for their newly proposed imputation method. (BN-Assisted 1-Nearest Neighbor Imputation: BNANNI, "Banana"?)

Previous Points:

Causality

I don’t know that the text added by the authors really helped. It could be perhaps condensed. As the authors have described, some people (famously including Pearl himself) consider the arrows of BNs to be “causal” while others observe that there’s nothing in the semantics requiring a causal interpretation. But as this discussion in the authors’ new text shows, if one is given a BN or one generates a BN, one could then assume the edges to be causal based on what comes down to personal preference. However, that assumption could then be true or false, for each edge, as the case may be. It doesn’t seem like calling edges of recreated biological Bayesian networks “causal” really gains anything – if the edges are between entities that are assumed to have a causal relationship based on current biological understanding, then those edges may well represent a causal link. And if not, then calling it “causal” is probably incorrect. Nothing is really gained in either case.

Including assumptions (1) (2) and (3) required for causality with no discussion of them is a little strange. For example, (1) isn’t a property of BNs in general. It’s a property of BNs that “accurately depict causality”, which is contingent on the domain being modeled, and a property that requires potentially difficult verification for general biological networks.

I would suggest just removing all references to causality in an effort to simplify and streamline this paper.

Throwing out whole samples.

It still seems to me that this could be remedied by adjusting equivalent assumed sample sizes using the priors; but I admit there’s no existing theoretical justification for this; placing this issue beyond the scope of the current investigation.

However, it doesn’t seem like throwing out whole samples is what’s typically done. Imputation of some sort seems much more likely to this reviewer. There’s no justification for this statement. Why is there no comparison to just (e.g.) imputing with the mean? Or any of the top 5 or six imputation methods? Most critically, imputation by nearest-neighbor is not new; there must be a comparison to existing nearest neighbor imputation methods.

Data double-dipping: (for evaluation of their imputation method on real datasets)

This still seems like a serious problem that has gone unaddressed. The authors are using their algorithm to perform imputation, then build a network, then generate data with the same missingness patterns as were addressed by imputation, then testing their imputation method against other paradigms based on how well a BN can be recreated based on the (imputed) data. This will systematically overestimate the performance of their imputation.

This system is clever as a method for evaluating accuracy in their framework; it just can’t start and end with using their own algorithm to evaluate their own algorithm’s accuracy.

Reviewer #2: I am happy that the reviewers have addressed my comments in the revised version, and the clarity of the paper has improved as a result. I have no further comments on the paper and am happy for it to be accepted

Reviewer #3: I do not see any major errors in this work that prevent publication. Concerns about how the paper is structured which pose a barrier to a reader understanding their work remain, but the authors have chosen not to address these.

**Have all data underlying the figures and results presented in the manuscript been provided?**

Reviewer #1: Yes

Reviewer #2: Yes

Reviewer #3: Yes

PLOS authors have the option to publish the peer review history of their article (what does this mean?). If published, this will include your full peer review and any attached files.

Reviewer #1: No

Reviewer #2: **Yes: **Jack Bowden

Reviewer #3: No

---

## [Editor Report · Decision Letter 2]

7 Sep 2021

Dear Dr Cordell,

We are pleased to inform you that your manuscript entitled "A Bayesian network approach incorporating imputation of missing data enables exploratory analysis of complex causal biological relationships" has been editorially accepted for publication in PLOS Genetics. Congratulations!

Yours sincerely,

Marco Scutari

Guest Editor

PLOS Genetics

David Balding

Section Editor: Methods

PLOS Genetics

Comments from the editors:

We believe that the authors addressed the remaining concerns of the reviewers. Therefore we recommend the publication of this submission in the current form and without further peer review. In particular, we appreciate that the authors have clarified that some of the concerns raised about the resubmission were outside of the central focus of the paper; and that they addressed the rest.

**Data Deposition**

http://datadryad.org/submit?journalID=pgenetics&manu=PGENETICS-D-21-00184R2

**Press Queries**

---

## [Editor Report · Acceptance letter]

24 Sep 2021

PGENETICS-D-21-00184R2 

A Bayesian network approach incorporating imputation of missing data enables exploratory analysis of complex causal biological relationships 

Dear Dr Cordell, 

We are pleased to inform you that your manuscript entitled "A Bayesian network approach incorporating imputation of missing data enables exploratory analysis of complex causal biological relationships" has been formally accepted for publication in PLOS Genetics! Your manuscript is now with our production department and you will be notified of the publication date in due course.

With kind regards,

Andrea Szabo

PLOS Genetics

On behalf of:
